# Spatially Invariant Unsupervised 3D Object-Centric Learning and Scene Decomposition

## Abstract

We tackle the problem of deep object-centric learning from a point cloud which is crucial for high-level relational reasoning and scalable machine intelligence. In particular, we introduce a framework, **SPAIR3D**, to factorize a 3D point cloud into a spatial mixture model where each component corresponds to one object. To model the spatial mixture model on point clouds, we derive the *Chamfer Mixture Loss*, which fits naturally into our variational training pipeline. Moreover, we adopt an object-specification scheme that describes each object's location relative to its local voxel grid cell. Such a scheme allows **SPAIR3D** to model scenes with an arbitrary number of objects. We evaluate our method on the task of unsupervised scene decomposition. Experimental results demonstrate that **SPAIR3D** has strong scalability and is capable of detecting and segmenting an unknown number of objects from a point cloud in an unsupervised manner.

## 1 Introduction

Motivated in part by cognitive psychology studies (Kahneman et al., 1992) that suggest human brains organize observations at an object level, recent advances in physical prediction (Chang et al., 2017) and reinforcement learning (Diuk et al., 2008; Kansky et al., 2017) have demonstrated superior robustness with environments that are modeled in an object-oriented manner. To exploit these techniques for 3D scenes, which can exhibit complex and combinatorially large observation spaces even when there are only a few basic elements, robust algorithms are needed to identity objects from 3D observations. We tackle in this paper the problem of deep object-centric learning from a point cloud, which is crucial for high-level relational reasoning and scalable machine intelligence.

There is a good body of existing literature on unsupervised object-centric generative models for images and videos. In particular, generative models based on Variational Autoencoders (VAE) (Kingma & Welling, 2014) have been employed to model the pixel intensities of an image with spatial Gaussian mixture models (Greff et al., 2019; Burgess et al., 2019; Lin et al., 2020; Crawford & Pineau, 2019; Eslami et al., 2016). SPAIR (Crawford & Pineau, 2019) makes use of an object-specification scheme which allows the method to scale well to scenes with a large number of objects.

The aforementioned works define objectness as a region with strong appearance correlations, and all of them employ VAE (Kingma & Welling, 2014; Higgins et al., 2017) in their structure. To be more precise, the encoder-decoder structure of VAE effectively creates an information bottleneck (Tishby et al., 1999; Burgess et al., 2018) limiting the amount of information passing through. To reconstruct the observation under limited information budget, highly correlated information must be exploited. Thus, VAE plays a critical role in exploiting such correlations. The above mentioned works mainly exploit appearance correlations on objects colored uniformly. In this work, we aim to prove that this paradigm is also applicable to structural correlations conveyed by point clouds without appearance information.

The irregularity of point cloud (detailed in section 3) renders a direct transfer of the above method infeasible. Inspired by SPAIR and the fact that 3D point cloud object generation requires centred-points (Chang et al., 2015), we propose in this paper a VAE-based model called **Sp**atially Invariant **A**ttend, **I**nfer, **R**epeat in **3D** (SPAIR3D), a model that generates spatial mixture distributions on point clouds to discover 3D objects in static scenes. Here we summarize the key contributions of this paper:

- We propose, to the best of our knowledge, the first unsupervised point cloud object-centric learning pipeline, named SPAIR3D.

- We also propose a new *Chamfer Mixture Loss* function tailored for learning mixture models over point cloud data with a novel graph neural network that can be used to model and generate a variable number of 3D points.

- We provide qualitative and quantitative results to show that SPAIR3D learns meaningful object-centric representation and decompose point clouds scene with an arbitrary number of objects in an object-oriented manner.

## 2    RELATED WORK

**Unsupervised Object-centric Learning.** Unsupervised object-centric learning has attracted increasing attention recently. A major focus of these methods is on joint object representation-learning and scene decomposition from images or videos via generative models (Burgess et al., 2019; Greff et al., 2019; Engelcke et al., 2020; Lin et al., 2020; Crawford & Pineau, 2019; Li et al., 2020; Chen et al., 2021). In particular, spatial Gaussian mixture models are commonly adopted to model pixel colors. The object-centric representation learning problem is then framed as a generative latent-variable modelling task. For example, IODINE (Greff et al., 2019) employs amortized inference that iteratively updates the latent representation of each object. GENESIS (Engelcke et al., 2020) and MONET (Burgess et al., 2019) sequentially decode each object. Slot attention (Locatello et al., 2020) and Neural Expectation Maximization (NEM) (Greff et al., 2017) also adopt the same formulation.

Instead of treating each component of the mixture model as a full-scale observation, **A**ttend, **I**nfer, **R**epeat (AIR) (Eslami et al., 2016) confines the extent of each object to a local region. To improve the scalability of AIR, SPAIR (Crawford & Pineau, 2019) employs a grid spatial attention mechanism to propose objects locally, which has proven effective in object-tracking tasks (Crawford & Pineau, 2020). To achieve a complete scene segmentation, SPACE (Lin et al., 2020) includes MONET in its framework for background modeling. Spatial attention models are also employed to reconstruct 3D scenes in the form of meshes or voxel in an object-centric fashion from a sequence of RGB frames (Henderson & Lampert, 2020). These methods similarly rely heavily on appearance and motion cues.

**Graph Neural Network for Point Cloud Generation.** Generative models such as VAEs (Gadelha et al., 2018) and generative adversarial networks (Achlioptas et al., 2018) have been successfully used for point-cloud generation but with a pre-defined number of points per object. Luo & Hu (2021) models the point cloud generation process as a latent variable conditioned Markov chain. Yang et al. (2019) proposed a VAE and normalizing flow-based approach that models object shapes as continuous distributions. While the proposed approach allows the generation of a variable number of points, it could not be naturally integrated into our framework because of the need for an ODE solver.

## 3    SPAIR3D

While the literature on deep unsupervised object-centric learning is rich, none of the methods listed above can be applied directly to 3D point cloud data. The reconstructions of 2D images, 3D voxel space, and neural radiance field in these methods are all coordinate-dependent (Watters et al., 2019; Stelzner et al., 2021). To be more precise, given a coordinate, an occupancy value (mask or density) and a feature vector (RGB color) are generated for each mixture component for that coordinate to form a Gaussian mixture model. For image data, the coordinate dependency can be implicitly embedded in the network structure since the input and output are of fixed sizes (Islam* et al., 2020). The coordinate thus provides the correspondence between input and reconstruction, inducing a likelihood function.

However, point cloud data are irregular and take the form of an unordered set. Each point cloud may have a varying number of points. Most importantly, the point coordinates carry all the structural information and form the reconstruction target, making coordinate-dependent reconstruction not an option. Due to the irregularity of point-cloud data, there is also usually no natural correspondence between the input and the reconstruction.While *Chamfer Distance* commonly serves as a loss function for point cloud reconstruction, it does not support mixture model formulation directly. Such data irregularity makes defining a mixture model over point cloud a non-trivial task. We, therefore, introduce **SPAIR3D**, a VAE-based generative model, to achieve 3D object-centric learning as well as

3D scene decomposition via object-centric point-cloud generation. It takes a point cloud as input and generates a structured latent representation for foreground objects and scene layout.

In the following, we first describe latent representation learning. We then leverage variational inference to jointly learn the generative model (§3.2) and inference model (§3.4). We further discuss the particular challenges arising for generative models in handling a varying number of points with a novel *Chamfer Mixture Loss* (§3.3) and *Point Graph Decoder* (§3.4).

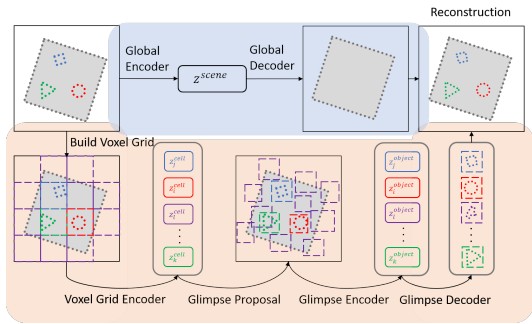
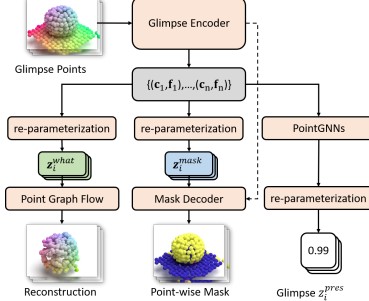

(a) Structure of SPAIR3D         (b) Structure of Glimpse VAE

Figure 1: (a) Structure of SPAIR3D. For better illustration, we adopt 2D abstraction and use colors to highlight important correspondence. (b) Structure of Glimpse VAE. Glimpse encoder encodes foreground glimpses and produce $\mathbf{z}_i^{what}$, $\mathbf{z}_i^{mask}$ and $z_i^{pres}$ for each glimpse. Point Graph Decoder takes $\mathbf{z}_i^{what}$ and reconstructs input points (left branch). Mask Decoder takes $\mathbf{z}_i^{mask}$ and generates masks for each point (middle branch). The dashed line represents the dependency on the coordinates of the intermediate points in the hierarchy and $\mathcal{G}_i$. Multi-layer PointGNN networks enable message passing between $(\mathbf{c}_i, \mathbf{f}_i)$ and produces $z_i^{pres}$ (right branch).

## 3.1 LOACL OBJECT PROPOSAL

As shown in Fig. 1a, SPAIR3D first divides a 3D scene into a spatial attention voxel grid. There can be empty voxel cells covering no points. We discard empty cells and associate a bounding box with each non-empty voxel cell. The set of input points captured by a bounding box is termed an *object glimpse*. Besides object glimpses, SPAIR3D also defines a *scene glimpse* covering all points in an input scene. Later, we show that we encode and decode (reconstruct) points in each glimpse and generate a mixing weight on each point to form a probability mixture model.

## 3.2 GENERATIVE MODEL

Similar to SPAIR, each grid cell generates posterior distributions over a set of latent variables defined as $\mathbf{z}_i^{cell} = \{\mathbf{z}_i^{where}, \mathbf{z}_i^{apothem}\}$, where $\mathbf{z}_i^{where} \in \mathbb{R}^3$ encodes the relative position of the center of the $i^{th}$ bounding box to the center of the $i^{th}$ cell, $\mathbf{z}_i^{apothem} \in \mathbb{R}^3$ encodes the apothem of the bounding box. Thus, each $\mathbf{z}_i^{cell}$ induces one object glimpse associated with the $i^{th}$ cell. Each object glimpse is then associated with posterior distributions over latent variables specified as $\mathbf{z}_i^{object} = \{\mathbf{z}_i^{what}, \mathbf{z}_i^{mask}, z_i^{pres}\}$, where $\mathbf{z}_i^{what} \in \mathbb{R}^A$ encodes the structure information of the corresponding object glimpse, $\mathbf{z}_i^{mask} \in \mathbb{R}^B$ encodes the mask for each point in the glimpse, $z_i^{pres} \in \{0, 1\}$ is a binary variable indicating whether the proposed object should exist ($z_i^{pres} = 1$) or not ($z_i^{pres} = 0$). The scene glimpse is associated with only one latent variable $\mathbf{z}^{scene} = \{\mathbf{z}_0^{what}\}$. We assume $z_i^{pres}$ follows a Bernoulli distribution. The posteriors and priors of other latent variables are all set to isotropic Gaussian distributions (see the appendix Sec. A for details).

Given latent representations of objects and the scene, the complete likelihood for a point cloud $\mathcal{X}$ is formulated as $p(\mathcal{X}) = \int_{\mathbf{z}} p(\mathbf{z})p(\mathcal{X}|\mathbf{z})d\mathbf{z}$, where $\mathbf{z} = \left(\bigcup_i \mathbf{z}_i^{cell}\right) \cup \left(\bigcup_i \mathbf{z}_i^{object}\right) \cup \mathbf{z}^{scene}$. As maximising the objective $p(\mathcal{X})$ is intractable, we resort to variational inference method to maximize its evidence lower bound (ELBO).

## 3.3 CHAMFER MIXTURE LOSS

Unlike generative model-based unsupervised 2D segmentation methods that reconstruct the pixel-wise appearance conditioning on its spatial coordinate, the reconstruction of a point cloud lost its point-wise

correspondence to the original point cloud. *Chamfer distance* is commonly adopted to measure the discrepancy between the generated point cloud ($\hat{\mathcal{X}}$) and the input point cloud ($\mathcal{X}$). Formally, *Chamfer distance* is defined by $d_{CD}(\mathcal{X}, \hat{\mathcal{X}}) = \sum_{x \in \mathcal{X}} \min_{\hat{x} \in \hat{\mathcal{X}}} \|x - \hat{x}\|_2^2 + \sum_{\hat{x} \in \hat{\mathcal{X}}} \min_{x \in \mathcal{X}} \|x - \hat{x}\|_2^2$. We refer to the first and the second term on the r.h.s as the forward loss and the backward loss, respectively.

Unfortunately, the *Chamfer distance* does not fit into the variational-inference framework. To get around that, we propose a *Chamfer Mixture Loss* (CML) tailored for training probability mixture models defined on point clouds. The *Chamfer Mixture Loss* is composed of a *forward likelihood* and a *backward regularization* corresponding to the forward and backward loss, respectively.

Denote the $i^{th}$ glimpse as $\mathcal{G}_i$, $i \in \{0, \dots, n\}$ and its reconstruction as $\hat{\mathcal{G}}_i$, $i \in \{0, \dots, n\}$. Specifically, we treat the scene glimpse as the $0^{th}$ glimpse that contains all input points, that is, $\mathcal{G}_0 = \mathcal{X}$. Note that one input point can be a member of multiple glimpses. Below we use $\mathcal{N}(\mu, \sigma)(x)$ to denote the probability density value of point $x$ evaluated at a Gaussian distribution of mean $\mu$ and variance $\sigma$. For each input point $x$ in the $i^{th}$ glimpse, the glimpse-wise forward likelihood of that point is defined as $\mathcal{L}_i^F(x) = \frac{1}{u_i} \max_{\hat{x} \in \hat{\mathcal{G}}_i} \mathcal{N}(\hat{x}, \sigma_c)(x)$, where $u_i = \int_{x \in \mathcal{X}} \max_{\hat{x} \in \hat{\mathcal{G}}_i} \mathcal{N}(\hat{x}, \sigma_c)(x) \mathrm{d}x$ is the normalizer and $\sigma_c$ is a hyperparameter. For each glimpse $\mathcal{G}_i$, $i \in \{0, \dots, n\}$, $\alpha_i^x \in [0, 1]$ defines a mixing weight for point $x$ in the glimpse and $\sum_{i=0}^n \alpha_i^x = 1$. In particular, $\alpha_i^x$, $i \in \{1, \dots, n\}$, is determined by $\alpha_i^x = \frac{z_i^{pres} \pi_i^x}{\sum_{j=1}^n z_j^{pres} \pi_j^x} z_i^{pres} \pi_i^x$, where $\pi_i^x$ is the predicted mask value and $\pi_i^x = 0$ if $x \notin \mathcal{G}_i$. The mixing weight for the scene layout points completes the distribution through $\alpha_0^x = 1 - \sum_{i=1}^n \alpha_i^x$ for $x \in \mathcal{G}_0$. Thus, the final mixture model for an input point $x$ is $\mathcal{L}^F(x) = \sum_{i=0}^n \alpha_i^x \mathcal{L}_i^F(x)$. The total forward likelihood of $\mathcal{X}$ is then defined as $\mathcal{L}^F(\mathcal{X}) = \prod_{x \in \mathcal{X}} \mathcal{L}^F(x)$.

The forward likelihood alone leads to a trivial sub-optimal solution with $\hat{\mathcal{X}}$ distributed densely and uniformly in the space. To enforce a high-quality reconstruction, we define a backward regularization term. For each predicted point $\hat{x}$, the point-wise backward regularization is defined as $\mathcal{L}^B(\hat{x}) = \max_{x \in \mathcal{G}_{i(\hat{x})}} \mathcal{N}(x, \sigma_c)(\hat{x})$, where $i(\hat{x})$ returns the glimpse index of $\hat{x}$. We denote $x(\hat{x}) = \arg\max_{x \in \mathcal{G}_{i(\hat{x})}} \mathcal{N}(x, \sigma_c)(\hat{x})$ and $\hat{\mathcal{X}} = \bigcup_{i=0}^n \hat{\mathcal{G}}_i$. The backward regularization is then defined as $\mathcal{L}^B(\hat{\mathcal{X}}) = \prod_{i=0}^n \prod_{\hat{x} \in \hat{\mathcal{G}}_i} \mathcal{L}^B(\hat{x})^{\alpha_i^{x(\hat{x})}}$. The exponential weighting, i.e. $\alpha_i^{x(\hat{x})} \in [0, 1]$, is crucial. As each predicted point $\hat{x} \in \hat{\mathcal{X}}$ belongs to one and only one glimpse, it is difficult to impose a mixture model interpretation on the backward regularization. The exponential weighting encourages the generated points in object glimpse to be close to input points with high probability belonging to $\mathcal{G}_i$. Combining the forward likelihood and the backward regularization together, we define *Chamfer Mixture Loss* as $\mathcal{L}_{\mathcal{CD}}(\mathcal{X}, \hat{\mathcal{X}}) = \mathcal{L}^F(\mathcal{X}) \cdot \mathcal{L}^B(\hat{\mathcal{X}})$. During inference, the segmentation label for each point $x$ is naturally obtained by $\arg\max_i \alpha_i^x$.

The overall loss function is $\mathcal{L} = -\log \mathcal{L}_{\mathcal{CD}}(\mathcal{X}, \hat{\mathcal{X}}) + \mathcal{L}_{KL}(\mathbf{z}^{cell}, \mathbf{z}^{object}, \mathbf{z}^{scene})$, where $\mathcal{L}_{KL}$ is the KL divergence between the prior and posterior of the latent variables (appendix Sec. A for details). In general, the normalizers in Chamfer Mixture Loss don't have closed-form solutions. However, the experiments below show that we can safely ignore the normalization constants during optimization.

## 3.4 MODEL STRUCTURE

We next introduce the encoder and decoder network structure for SPAIR3D. The building blocks are based on graph neural networks and point convolution operations (See appendix Sec. C for details).

**Encoder network.** We design an encoder network $q_\phi(\mathbf{z}|x)$ to obtain the latent representations $\{\mathbf{z}_i^{cell}\}_{i=1}^n$ and $\{\mathbf{z}_i^{object}\}_{i=1}^n$ from a point cloud. To achieve the spatially invariant property, we group one PointConv (Wu et al., 2019) layer and one PointGNN (Shi & Rajkumar, 2020) layer into pairs for message passing and information aggregation among points and between cells.

**(a) Voxel Grid Encoding**. The voxel-grid encoder takes a point cloud as input and generates for each spatial attention voxel cell $\mathcal{C}_i$ two latent variables $\mathbf{z}_i^{where} \in \mathbb{R}^3$ and $\mathbf{z}_i^{apothem} \in \mathbb{R}^3$ to propose a glimpse $\mathcal{G}_i$ potentially occupied by an object.

To better capture the point cloud information in $\mathcal{C}_i$, we build a voxel pyramid within each cell $\mathcal{C}_i$ with the bottom level corresponding to the finest voxel grid. We aggregate information hierarchically using PointConv-PointGNN pairs from bottom to top through each level of the pyramid. For each layer of the pyramid, we aggregate the features of all points and assign it to the point spawned at the

center of mass of the voxel cell. Then PointGNN is employed to perform message passing on the radius graph built on all spawned points. The output of the final aggregation block produces $\mathbf{z}_i^{where}$ and $\mathbf{z}_i^{apothem}$ via the re-parametrization trick (Kingma & Welling, 2014).

We obtain the offset distance of a glimpse center from its corresponding grid cell center using $\Delta g_i = \tanh(\mathbf{z}_i^{where}) \cdot L$, where $L$ is the maximum offset distance. The apothems of the glimpse in the $x, y, z$ direction is given by $\Delta \mathbf{g}_i^{apo} = T(\mathbf{z}_i^{apothem})(\mathbf{r}^{max} - \mathbf{r}^{min}) + \mathbf{r}^{min}$, where $T(\cdot)$ is the sigmoid function and $[\mathbf{r}^{min}, \mathbf{r}^{max}]$ defines the range of apothem.

**(b) Glimpse Encoding.** Given the predicted glimpse center offset and the apothems, we can associate one glimpse with each spatial attention voxel cell. We adopt the same encoder structure to encode each glimpse $\mathcal{G}_i$ into one point $\mathbf{a}_i = (\mathbf{c}_i, \mathbf{f}_i)$, where $\mathbf{c}_i$ is the glimpse center coordinate and $\mathbf{f}_i$ is the glimpse feature vector. We then generate $\mathbf{z}_i^{what}$ and $\mathbf{z}_i^{mask}$ from $\mathbf{a}_i$ via the re-parameterization trick.

The generation of $z_i^{pres}$ determines the glimpse rejection process and is crucial to the final decomposition quality. Unlike previous work (Crawford & Pineau, 2019; Lin et al., 2020), SPAIR3D generates $z^{pres}$ from glimpse features instead of cell features based on our observation that message passing across glimpses provides more benefits in the glimpse-rejection process. To this end, a radius graph is first built on the point set $\{(\mathbf{c}_i, \mathbf{f}_i)\}_{i=1}^n$ to connect nearby glimpse centers, which is followed by multiple PointGNN layers with decreasing output channels to perform local message passing. The $z_i^{pres}$ of each glimpse is then obtained via the re-parameterization trick. Information exchange between nearby glimpses can help avoid over-segmentation that would otherwise occur because of the high dimensionality of point cloud data.

**(c) Global Encoding.** The global encoding module adopts the same encoder as the object glimpse encoder to encode scene glimpse $\mathcal{G}_0$. The learned latent representation is $\mathbf{z}_0^{what}$ with $z_0^{pres} = 1$.

**Decoder network.** We now introduce the decoders used for point-cloud and mask generation.

**(a) Point Graph Decoder (PGD).** Given the $\mathbf{z}_i^{object}$ of each glimpse, the decoder is used for point-cloud reconstruction as well as segmentation-mask generation. In reconstruction, the number of generated points has a direct effect on the magnitudes of the forward and backward terms in the Chamfer Mixture Loss. An unbalanced number of reconstruction points can lead to under- or over-segmentation. To balance the forward likelihood and the backward regularization, the number of predictions for each glimpse should be approximately the same as the number of input points. We propose a graph network based point decoder to allow setting the size of $\hat{\mathcal{X}}$ in run time.

Similar to (Luo & Hu, 2021), we treat the point cloud reconstruction as a point diffusion process. The input to the PGD is a set of 3D points with coordinates sampled from a zero-centered Gaussian distribution, with the population determined by the number of points in the current glimpse. Features of the input points are set uniformly to the latent variable $\mathbf{z}_i^{what}$. PGD is composed of several PointGNN layers, each of which is preceded by a radius graph operation. The output of each PointGNN layer is of dimension $f + 3$, with the first $f$ dimensions interpreted as the updated features and the last 3 dimensions interpreted as the updated 3D coordinates for estimated points. Since we only focus on point coordinates prediction, we set $f = 0$ for the last PointGNN layer.

**(b) Mask Decoder.** The Mask Decoder decodes $(\mathbf{c}_i, \mathbf{z}_i^{mask})$ to the mask value, $\pi_i^x \in [0, 1]$, of each point within a glimpse $\mathcal{G}_i$. The decoding process follows the exact inverse pyramid structure of the Glimpse Encoder. To be more precise, the mask decoder can access the spatial coordinates of the intermediate aggregation points of the Glimpse Encoder as well as the point coordinates of $\mathcal{G}_i$. During decoding, PointConv is used as deconvolution operation.

**Glimpse VAE and Global VAE.** The complete Glimpse VAE structure is presented in Fig. 1b. The Glimpse VAE is composed of a Glimpse Encoder, Point Graph Decoder, Mask Decoder and a multi-layer PointGNN network. The Glimpse Encoder takes all glimpses as input and encodes each glimpse $\mathcal{G}_i$ individually and in parallel into feature points $(\mathbf{c}_i, \mathbf{f}_i)$. Via the re-parameterization trick, $\mathbf{z}_i^{what}$ and $\mathbf{z}_i^{mask}$ are then obtained from $\mathbf{f}_i$. From there, we use the Point Graph Decoder to decode $\mathbf{z}_i^{what}$ to reconstruct the input points, and we use the Mask Decoder to decode $\mathbf{z}_i^{mask}$ to assign a mask value for each input point within $\mathcal{G}_i$. Finally, $\mathbf{z}_i^{pres}$ is generated using message passing among neighbour glimpses. The processing of all glimpses happens in parallel. The Global VAE consisting of the Global Encoder and a PGD outputs the reconstructed scene layout.

### 3.5 SOFT BOUNDARY

The prior of $\mathbf{z}^{apothem}$ is set to encourage apothem to shrink so that the size of the glimpses will not be overly large. However, if points are excluded from one glimpse, the gradient from the likelihood of the excluded points will not influence the size and location of the glimpse anymore, and this can lead to over-segmentation. To solve this problem, we introduce a soft boundary weight $b_i^x \in [0, 1]$ which decreases when a point $x \in \mathcal{G}_i$ moves away from the bounding box of $\mathcal{G}_i$. Taking $b_i^x$ into the computation of $\alpha$, we obtain an updated mixing weight $\overline{\alpha}_i^x = \frac{z_i^{pres} \pi_i^x b_i^x}{\sum_{j=1}^n z_j^{pres} \pi_j^x b_j^x} z_i^{pres} \pi_i^x b_i^x$. By employing such a boundary loss, the gradual exclusion of points from glimpses will be reflected in gradients to counter over-segmentation. Details can be found in appendix Sec. B.

## 4 EXPERIMENTS

### 4.1 DATASETS

While many benchmark datasets are established (Johnson et al., 2017; Kabra et al., 2019) for unsupervised object-centric learning, they don't provide camera poses and depth images, which are necessary for point cloud generation. Thus, we introduce two new point-cloud datasets *Unity Object Room* and *Unity Object Table* built on the *Unity* platform (Juliani et al., 2020). The Unity Object Room (UOR) dataset is built to approximate Object Room (Kabra et al., 2019) dataset but with increased scope and complexity. In each scene, objects sampled from a list of 8 regular geometries are randomly placed on a square floor. The Unity Object Table (UOT) dataset approximates Robotic Object Grasping scenario where multiple objects are placed on a round table. Instead of using objects of simple geometries, we create each scene of the UOT dataset with more challenging objects selected from a pool of 9 irregular objects. For both datasets, the number of objects placed in each scene varies from 2 to 5 with equal probabilities. During the scene generation, the size and orientation of the objects are varied randomly within a pre-defined range.

We capture the depth, RGB, normal frames, and pixel-wise semantics as well as instance labels for each scene from 10 different viewpoints. This setup aims to approximate the scenario where a robot equipped with depth and RGB sensors navigates around target objects and captures data. The point cloud data for each scene is then constructed by merging these 10 depth maps. For each dataset, we collect $50K$ training scenes, $10K$ validation scenes and $5K$ testing scenes. In-depth dataset specification and analysis can be found in supplementary D.

### 4.2 UNSUPERVISED SEGMENTATION

**Baseline.** Due to the sparse unsupervised 3D point cloud object-centric learning literature, we could not find a generative baseline to compare with. Thus, we compare SPAIR3D with PointGroup (PG) (Jiang et al., 2020), a recent supervised 3D point cloud segmentation model. PointGroup performs semantic prediction and instance predictions from a point cloud and RGB data using a model trained with ground-truth semantic labels and instance labels. To ensure a fair comparison, we assign each point the same color (white) so appearance information doesn't play a role. The PointGroup network is fine-tuned on the validation set to achieve the best performance.

**Performance Metric.** We use the Adjust Rand Index (ARI) (Hubert & Arabie, 1985) to measure the segmentation performance against the ground truth instance labels. We also employ foreground Segmentation Covering (SC) and foreground unweighted mean Segmentation Covering (mSC) (Engelcke et al., 2020) for performance measurements as ARI does not penalize models for object over-segmentation (Engelcke et al., 2020).

**Evaluation.** Table 1 shows that SPAIR3D achieves comparable performance to the supervised baseline on both UOT and UOR datasets. As demonstrated in Fig. 2, each foreground object is proposed by one and only one glimpse. The scene layout is separated from objects and accurately modelled by the global VAE. It is worth noting that the segmentation errors mainly happen at the bottom of objects. Without appearance information, points at the bottom of objects are also correlated to the ground. In Fig. 4, we sort the test data based on their performance in an ascending order and plot the performance distributions. As expected, the supervised baseline (Orange) performs better but SPAIR3D manages to achieve high-quality segmentation (SC score $> 0.8$) on around $80\%$ of the scenes without supervision. See appendix Sec. E for more segmentation results including failure cases.

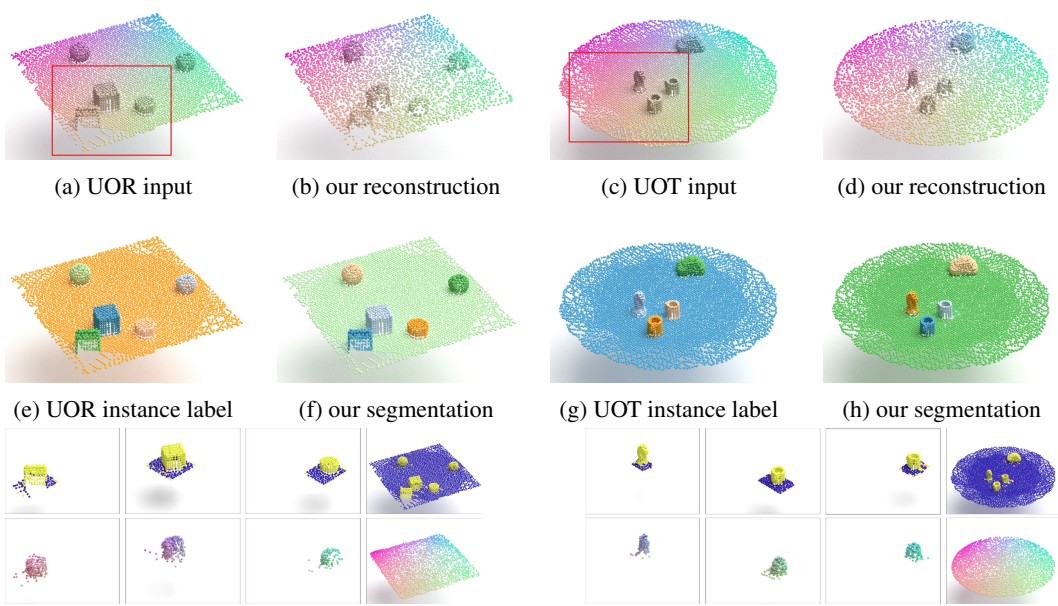

(a) UOR input    (b) our reconstruction    (c) UOT input    (d) our reconstruction

(e) UOR instance label    (f) our segmentation    (g) UOT instance label    (h) our segmentation

(i) UOR close-up glimpses visualization, foreground alpha and scene layout reconstruction.

(j) UOT close-up glimpses visualization, foreground alpha and scene layout reconstruction.

Figure 2: Visualization of segmentation results on UOR and UOT dataset.

Table 1: 3D point cloud segmentation results on UOR dataset (blue) and UOT dataset (red).

| UOR / UOT | ARI↑ | SC↑ | mSC↑ |
|---|---|---|---|
| PG | 0.976 | 0.907 | 0.900 |
|  | 0.923 | 0.917 | 0.907 |
| Ours | $0.915 \pm 0.03$ | $0.832 \pm 0.04$ | $0.836 \pm 0.04$ |
|  | $0.901 \pm 0.02$ | $0.835 \pm 0.03$ | $0.831 \pm 0.03$ |
| voxel size $0.75l$ | 0.932 | 0.853 | 0.850 |
| voxel size $1.25l$ | 0.922 | 0.857 | 0.861 |
| $6-12$ objects | 0.912 | 0.846 | 0.842 |
|  | 0.892 | 0.843 | 0.834 |
| object matrix | 0.872 | 0.856 | 0.861 |
|  | 0.879 | 0.877 | 0.886 |

It is worth noting that UOR and UOT datasets include around $25\%$ objects (for scenes of 2 to 5 objects) and $60\%$ objects (for scenes of 6 to 12 objects) that are close to its neighbors with touching or almost touching surfaces. The reported quantitative (Table 1) and qualitative results (Fig. 5(a)–(d)) show that our method achieves stable performance for those challenging scenes.

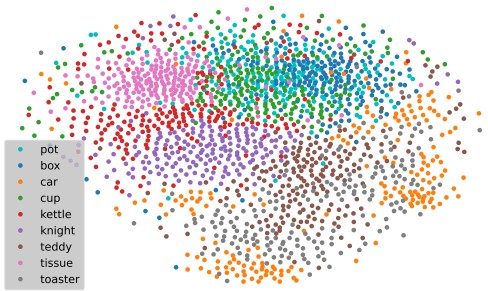

pot
box
car
cup
kettle
knight
teddy
tissue
toaster

Figure 3: t-SNE visualization of $z^{what}$ on UOT

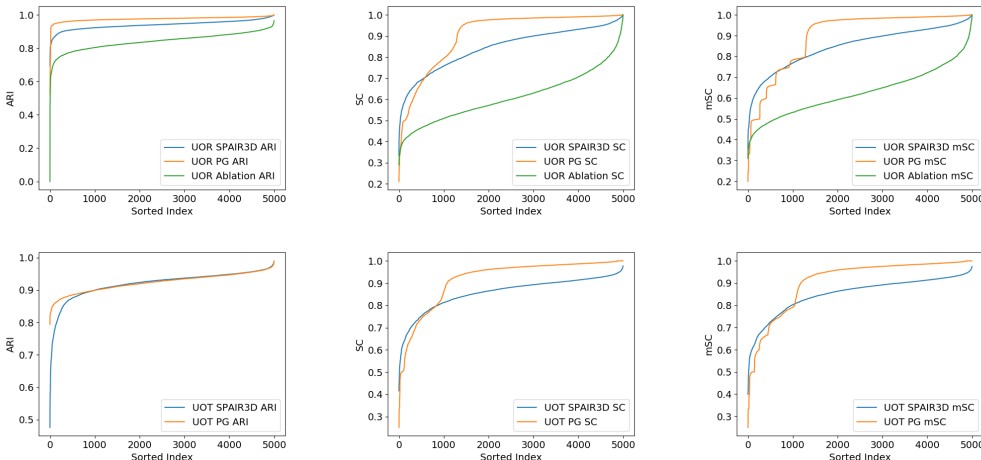

Figure 4: Test set performance distributions on UOR (first row) and UOT (second row).

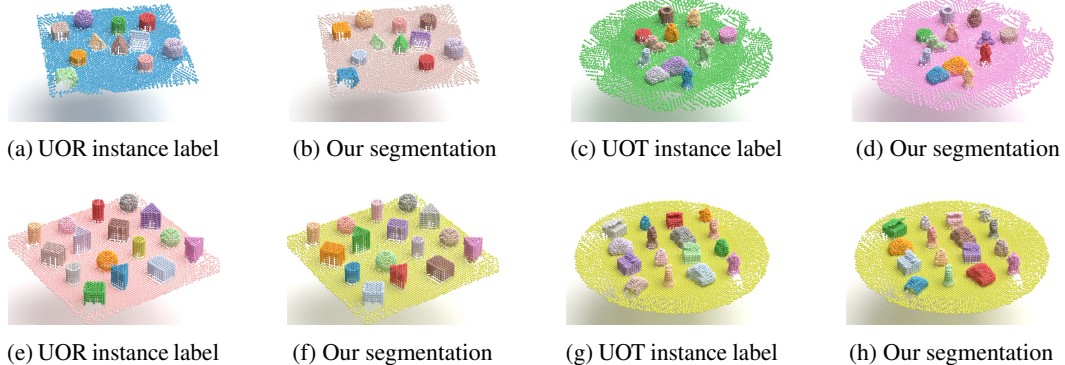

(a) UOR instance label    (b) Our segmentation    (c) UOT instance label    (d) Our segmentation

(e) UOR instance label    (f) Our segmentation    (g) UOT instance label    (h) Our segmentation

Figure 5: Segmentation on scenes with 6 to 12 objects (a - d) and on object matrix (e-h)

### 4.3 OBJECT CENTRIC REPRESENTATION

To show that our model learns meaningful representations, for each object type in UOT dataset we collect the $z^{what}$ of 200 instances and visualize them with t-SNE algorithm (van der Maaten & Hinton, 2008). Fig 3 clearly shows the $z^{what}$ of different object types cluster at different regions. Not surprisingly, the embeddings of *pot* and *box* instances occupy the same area since they have almost identical spatial structure. See appendix Sec. D for object specification.

### 4.4 VOXEL SIZE ROBUSTNESS AND SCALABILITY

In the literature (Lin et al., 2020; Crawford & Pineau, 2019), the cell voxel size, an important hyperparameter, is chosen to match the object size in the scene. To evaluate the robustness of our method w.r.t voxel size, we train our model on the UOR dataset with voxel size set to $0.75l$ and $1.25l$ with $l$ being the average size of the objects. Results in Table 1 show that our method achieves the stable performance w.r.t the voxel size.

To demonstrate scalability, we evaluate our pre-trained model on 1000 scenes containing $6 - 12$ randomly selected objects and report performance in Table 1. Due to the spatial invariance property, SPAIR3D suffers no performance drop on $6 - 12$ object scenes that were never used for training.

We also evaluated our approach on scenes termed as *Object Matrix*, which consists of 16 objects placed in a matrix form. We fixed the position of all 16 objects but set their size and rotation randomly. For each dataset, SPAIR3D is evaluated on 100 *Object Matrix* scenes. The results are reported in Table 1. Note that our model is trained on scenes with 2 to 5 objects, which is less than one-third of the number of objects in *Object Matrix* scenes. Fig 5(e)-(h) is illustrative of the results.

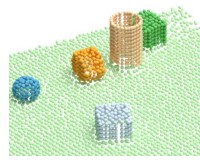 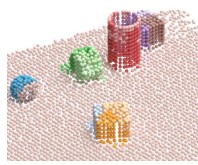

(a) With PointGNNs     (b) w/o PointGNNs

Figure 6: The comparison between models with (left) and without (right) multi-layer PointGNNs. It shows that objects are over-segmented severely without multi-layer PointGNNs.

### 4.5 Ablation Study of Multi-layer PointGNN

To evaluate the importance of multi-layer PointGNN in $\mathbf{z}_i^{pres}$ generation (right branch in Fig. 1b), we remove the multi-layer PointGNN and generate $\mathbf{z}_i^{pres}$ directly from $\mathbf{f}_i$. The ablated model on the UOR dataset achieves **ARI:0.841**, **SC:0.610**, and **mSC:0.627**, which is significantly worse than the full SPAIR3D model. The performance distribution of ablated SPAIR3D (Fig 4, first row) indicates that removing the multi-layer PointGNN has a negative influence on the entire dataset. Fig. 6 shows that the multi-layer PointGNN is crucial to preventing over-segmentation.

### 4.6 Empirical Evaluation of PGD

3D objects of the same category can be modeled by a varying number of points. The generation quality of the point cloud largely depends on the robustness of our model against the number of points representing each object. To demonstrate that PGD can reconstruct each object with a dynamic number of points, we train the global VAE on the ShapeNet dataset (Chang et al., 2015), where each object is composed of roughly 2000 points, and reconstruct the object with a varying number of points. For reference input point clouds of size $N$, we force PGD to reconstruct a point cloud of size $1.5N$, $1.25N$, $N$, $0.75N$, and $0.5N$, respectively. As shown in Fig. 7, while with less details compared to the input, the reconstructions capture the overall object structure in all 5 settings.

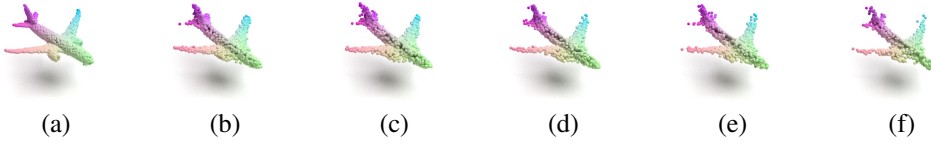

(a)     (b)     (c)     (d)     (e)     (f)

Figure 7: PGD trained on ShapeNet. (a) Input point cloud with $N$ points. Reconstruction with (b) $1.5N$, (c) $1.25N$, (d) $N$, (e) $0.75N$, and (f) $0.5N$ points.

## 5 Limitations

SPAIR3D extends the SPAIR framework to point cloud data and inherits its strengths and limitations. Similar to SPAIR, the key to achieving high scalability in SPAIR3D is the independence assumption among objects, which is reflected in the local attention and reconstruction mechanisms. The price we pay for that scalability is that the local object proposal mechanism can only reliably detect objects whose size fall within a bounded range. By design, each voxel cell can only propose one object. Thus, it is difficult to detect multiple objects that exist in the same voxel cell. If one object is much larger than the size of the voxel cells, no voxel cells can accurately infer complete object information from its local perceptive field. Given these limitations are inherent to the SPAIR framework, they can really only be resolved at a meta level. In our intended use case, an intelligent agent that employs SPAIR3D as a subroutine can use voxel cells of vastly different sizes to segment out objects of different sizes and use a spatial knowledge representation and reasoning formalism [Cohn & Renz (2008)] to resolve inconsistencies and get to a full scene representation.

## 6 Conclusion and Future Work

We propose *SPAIR3D*, to the best of our knowledge, the first generative unsupervised object-centric learning model on point cloud with applications to 3D object segmentation task. Based on experimental results on UOR and UOT datasets, we demonstrate that SPAIR3D can generalise well to previously unseen scenes with a large number of objects without performance degeneration. The spatial mixture interpretation of SPAIR3D allows the integration of memory mechanism (Bornschein et al., 2017) or iterative refinement (Greff et al., 2019). Our model may tend to over-segment objects of a significant different scale from those in the training dataset, which is left as our future work.

## 7 ETHICAL STATEMENTS

Similar to most data-driven approaches, the proposed method also potentially brings the risk of learning biases. While our object-centric representation learning has shown good generalisation ability to new scenes, the application of our method should be done with careful consideration of the consequences from any potential underlying biases in the data-collection process.

## 8 REPRODUCIBILITY STATEMENT

We described in details our model structure, the choice of hyperparameter in appendix Sec. A, B, and C. Dataset construction details are presented in appendix Sec D. We believe our framework can be reproduced based on these information.

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

## A  HYPERPARAMETERS AND PRIOR DISTRIBUTIONS

In this section, we present hyperparameters we used in our model as well as definitions of the prior distributions of latent variables. We use tuples in the form of $(n, m, p, q)$ to denote annealing of hyperparameter values from $n$ to $m$, starting from iteration $p$ to iteration $q$. Glimpse related hyperparameters are shown in Table 2. Priors are specified in Table 3. Other hyperparameters are specified in Table 4.

Table 2: Glimpse related hyperparameters.

| Term | Value |
|---|---|
| spatial attention grid cell size | 1 |
| glimpse max apothem | 1 |
| glimpse min apothem | 0.25 |
| glimpse max center offset | 0.75 |

Table 3: Prior distributions.

| Term | Value |
|---|---|
| $\mu^{apothem}$ | $(2, -1, 10000, 20000)$ |
| $\beta^{pres}$ | $(0.01, 0.0001, 0, 15000)$ |
| $z^{where}$ prior | $\mathcal{N}(0, 0.5)$ |
| $z^{apothem}$ prior | $\mathcal{N}(\mu^{apothem}, 0.5)$ |
| $z^{pres}$ prior | $Bernoulli(\beta^{pres})$ |
| $z^{what}$ prior | $\mathcal{N}(0, 1)$ |
| $z^{mask}$ prior | $\mathcal{N}(0, 1)$ |

Table 4: Other hyperparameters.

| Term | Value |
|---|---|
| CML $\sigma_c$ | $(0.1, 0.05, 10000, 15000)$ |
| CML dist. | $\mathcal{N}(0, \sigma_c)$ |
| relaxed Bernoulli temp | $(2.5, 0.5, 0, 10000)$ |
| PGD initial distribution | $\mathcal{N}(0, 0.3)$ |

**Mixing Weight.** Recall that the mixing weight defined in Eq. (6) in our main paper for the *forward Chamfer Likelihood* implies the segmentation of the point cloud. To encourage the mixing weight to approach 0 or 1, we include a temperature factor 10 into the computation of $\alpha_i^x$, and the mixing weight is implemented as

$$\alpha_i^x = \frac{(z_i^{pres}\pi_i^x)^{10}}{\sum_{j=1}^n (z_j^{pres}\pi_j^x)^{10}} z_i^{pres}\pi_i^x. \tag{1}$$

**Weight for Regularisation with Priors.** Here we introduce the detailed formulation of $\mathcal{L}_{KL}$, the second term of the evidence lower bound, in the following Eq. 2.

$$\begin{aligned}
\mathcal{L}_{KL}(\mathbf{z}^{cell}, \mathbf{z}^{object}, \mathbf{z}^{scene}) = w\mathcal{D}_{KL}(p(z^{pres})||q(z^{pres}|x)) + z^{pres}[ \\
\mathcal{D}_{KL}(p(\mathbf{z}^{what})||q(\mathbf{z}^{what}|x)) + \\
\mathcal{D}_{KL}(p(\mathbf{z}^{mask})||q(\mathbf{z}^{mask}|x)) + \\
\mathcal{D}_{KL}(p(\mathbf{z}^{where})||q(\mathbf{z}^{where}|x)) + \\
\mathcal{D}_{KL}(p(\mathbf{z}^{apothem})||q(\mathbf{z}^{apothem}|x))].
\end{aligned} \tag{2}$$

Note that the weight $w$ for KL Divergence of $z^{pres}$ is to encourage glimpse rejection. More specifically, $w$ is set to 10 for UOR dataset and is annealed from 10 to 20 in the first 15000 steps for UOT dataset in our experiments. Following Crawford & Pineau (2019), we also set the weight for KL divergence of the rest latent variables with $z^{pres}$ so that the rejected glimpses won't produce penalties encouraging glimpse rejection.

**Training.** We use Adam (Kingma & Ba, 2014) optimizer with learning rate set to 0.0001 during our training process. Training takes roughly 4 days on one RTX 3090 for both datasets.

## B    SOFT BOUNDARY

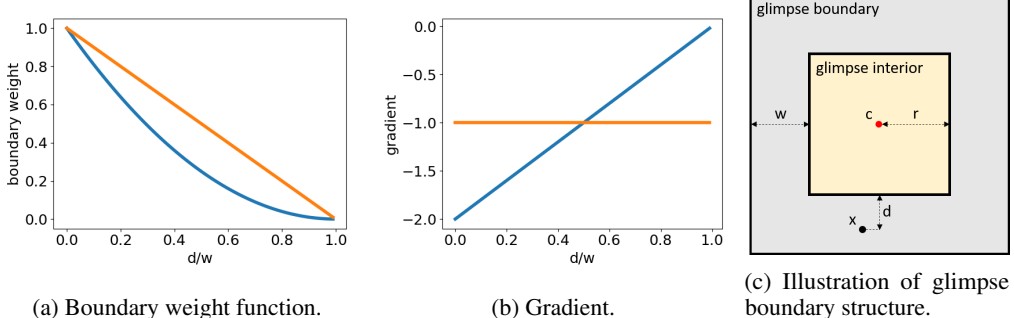

(a) Boundary weight function.    (b) Gradient.    (c) Illustration of glimpse boundary structure.

Figure 8: Visualization of glimpse boundary structure, glimpse boundary weights and the corresponding gradients. Fig. 8c illustrates the glimpse structure where $c$ is the glimpse center and $x$ is an point living in the glimpse boundary. The linear decay function (orange) and parabola decay function (blue) are plotted in Fig. 8a with the corresponding gradients shown in Fig. 8b.

As shown in Fig 8c, we divide a glimpse into glimpse interior of apothem $r$ and glimpse boundary of width $w$. For an input point $x$ in glimpse $\mathcal{G}_i$ with center location $c_i$, its boundary weight $b_i^x \in (0, 1]$ is defined as

$$b_i^x = \begin{cases} 1, & x \in \text{glimpse interior}; \\ f(d, w), & x \in \text{glimpse boundary}, \end{cases}$$

where $f$ is a continuous function and $d = \|x - c_i - r\|_{\text{inf}}$.

One example of $f$ is a linear decay function $f(d, w) = 1 - \frac{d}{w}$. Another example is a parabola decay function $f(d, w) = (\frac{d}{w})^2 - 2(\frac{d}{w}) + 1$, which generates a larger gradient compared with the linear one when $d \leq \frac{w}{2}$, and a smaller gradient, otherwise. For the case where the distance between two objects is smaller than $w$ but larger than $\frac{w}{2}$, parabola decay function is preferred to preventing overly large glimpse. In this work, we use parabola decay function and set the width of the glimpse boundary as $w = 0.75r$.

From Fig. 8a and Fig. 8b, we observe that both types of boundary decay function $f$ provide a negative gradient when points are being excluded from a glimpse which has increasing $\frac{d}{w}$ value.

## C    MODEL STRUCTURE

In this section, we introduce the detailed model structure. We first introduce the notation that we used in our network structure. One PointConv layer is specified with $(c_{mid}, c_{out})$ (Wu et al., 2019) (See Fig. 9 for structure illustration). To achieve spatially invariant operation in PointConv, we divide the feature value of each point by the total number of points involved for each point convolution operation instead of the pre-computed point density as in (Wu et al., 2019).

One PointGNN layer contains three two-layer MLPs which are $MLP_h$, $MLP_f$ and $MLP_g$, respectively (Shi & Rajkumar, 2020) and the structure can be summarized by

$$s_i^{out} = g(\max_{j \in \mathcal{N}_i}\{f(x_j - x_i + h(s_i^{in}), s_i^{in})\}, s_i^{in}), \tag{3}$$

where $s_i^{in}$ and $s_i^{out}$ are the features of the point $i$ before and after PointGNN layer, $\mathcal{N}_i$ defines the points connected to $i$, $x_i$ indicates the 3D coordinate of the point $i$. The max operation is performed over all points $j$ that are connected to the point $i$. For all PointGNN layers, we consistently set $h_{hidden} = 32$ and $h_{out} = 3$. Thus, we define the structure of a PointGNN layer with parameters as $(f_{hidden}, f_{out}, g_{hidden}, g_{out})$. We also list the parameters for other operations such as the radius

for radius graph operation, the voxel cell size for voxel pooling operation, and the output size for linear layers in following tables. Table 5 shows the structure of voxel grid encoder. The structure of glimpse VAE including the glimpse encoder, the mask decoder, the glimpse Point Graph Flow, and the Multi-layer PointGNN is presented in Table 6, Table 9, Table 8 and Table 11. The structure of global VAE including the Global encoder and the Global Point Graph Flow is detailed in Table 10 and Table 7. Input point coordinates are reduced by a factor of 16.

Table 5: Voxel grid encoder.

| Layer/Operation | Parameter |
|---|---|
| Radius Graph | 0.0625 |
| PointConv | $(8, 8)$ |
| Celu | |
| PointConv | $(16, 16)$ |
| Celu | |
| PointConv | $(32, 32)$ |
| Celu | |
| Voxel Pool | 0.03125 |
| PointConv | $(32, 64)$ |
| Celu | |
| Radius Graph | 0.03125 |
| PointGNN | $(64, 64, 64, 64)$ |
| LayerNorm | |
| Voxel Pool | 0.0625 |
| PointConv | $(64, 128)$ |
| Celu | |
| Radius Graph | 0.125 |
| PointGNN | $(128, 128, 128, 128)$ |
| LayerNorm | |
| Voxel Pool | 0.125 |
| PointConv | $(128, 256)$ |
| Celu | |
| Radius Graph | 0.25 |
| PointGNN | $(256, 256, 256, 256)$ |
| PointGNN | $(256, 256, 256, 256)$ |
| PointGNN | $(256, 256, 256, 256)$ |
| LayerNorm | |
| PointConv | $(256, 256)$ |
| Linear | 12 |

Table 6: Glimpse encoder.

| Layer/Operation | Parameter |
|---|---|
| Radius Graph | 0.25 |
| PointGNN | $(8, 8, 8, 8)$ |
| LayerNorm | |
| Voxel Pool | 0.25 |
| PointConv | $(16, 32)$ |
| Celu | |
| Radius Graph | 0.5 |
| PointGNN | $(32, 32, 32, 32)$ |
| LayerNorm | |
| Voxel Pool | 0.25 |
| PointConv | $(64, 128)$ |
| Celu | |
| Radius Graph | 1.0 |
| PointGNN | $(128, 128, 128, 128)$ |
| LayerNorm | |
| PointConv | $(128, 256)$ |
| Celu | |
| Linear | 256 |

Table 7: Global Point Graph Flow.

| Layer/Operation | Parameter |
|---|---|
| Random Sampling | |
| Radius Graph | 0.2 |
| PointGNN | $(128, 128, 128, 64 + 3)$ |
| Radius Graph | 0.1 |
| PointGNN | $(64, 64, 64, 32 + 3)$ |
| Radius Graph | 0.05 |
| PointGNN | $(16, 16, 16, 3)$ |

Table 8: Glimpse Point Graph Flow.

| Layer/Operation | Parameter |
|---|---|
| Random Sampling | |
| Radius Graph | 0.2 |
| PointGNN | $(128, 128, 128, 64 + 3)$ |
| Radius Graph | 0.1 |
| PointGNN | $(64, 64, 64, 32 + 3)$ |
| Radius Graph | 0.05 |
| PointGNN | $(16, 16, 16, 3)$ |

Table 9: Mask decoder.

| Layer/Operation | Parameter |
|---|---|
| PointConv | $(64, 32)$ |
| Celu | |
| PointConv | $(16, 16)$ |
| Celu | |
| PointConv | $(8, 8)$ |
| Celu | |
| Linear | 1 |

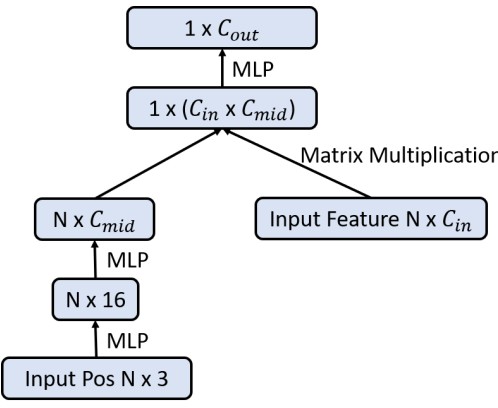

Figure 9: Structure of PointConv.

Table 10: Global encoder.

| Layer/Operation | Parameter |
|---|---|
| Radius Graph | 0.25 |
| PointGNN | $(8, 8, 8, 8)$ |
| LayerNorm | |
| Voxel Pool | 0.25 |
| PointConv | $(16, 32)$ |
| Celu | |
| Radius Graph | 0.5 |
| PointGNN | $(32, 32, 32, 32)$ |
| LayerNorm | |
| Voxel Pool | 0.25 |
| PointConv | $(64, 128)$ |
| Celu | |
| Radius Graph | 1.0 |
| PointGNN | $(128, 128, 128, 128)$ |
| LayerNorm | |
| PointConv | $(128, 256)$ |
| Celu | |
| PointConv | $(256, 512)$ |

Table 11: Multi-layer PointGNN.

| Layer/Operation | Parameter |
|---|---|
| Random Sampling | |
| Radius Graph | 1.0 |
| PointGNN | $(128, 64, 64, 64)$ |
| PointGNN | $(32, 32, 32, 32)$ |
| PointGNN | $(16, 16, 16, 8)$ |
| Linear | 1 |

## D  DATASET SPEC

We provide more details about the UOR and UOT dataset in this section. For both datasets, in each scene, 2-5 objects are uniformly randomly selected (with replacement) from the candidate set and placed at random locations in the scene with the constraint that they cannot largely overlap with each other (slight overlapping and touching are permitted). All objects are randomly rotated along y-axis.

For the point cloud construction, we convert the 10 depth frames for each scene into 10 partial point clouds. To down sample the point clouds, we apply voxel grid pooling on the 10 partial point clouds. More specifically, a voxel grid with cell size 0.15 is placed across the space where each partial point cloud exists. For all points in one cell, the coordinates are aggregated by average pooling operation. After we merge the 10 partial point clouds into one, we perform voxel grid pooling again to get the final input point cloud. We divide the coordinates of all points by 8 to scale the complete point clouds.

The intrinsic parameters of cameras are shown in Table 12, which is crucial to accurate point cloud construction. The object pool for UOR and UOT are presented in Table 13 and Table 14, respectively, where we also specify the detailed dimension range of each object.

Table 12: Camera intrinsic parameters

| Term | Value |
| --- | --- |
| focal length | 10 mm |
| sensor size x | 16 mm |
| sensor size y | 16 mm |
| clipping plane | 20 m |

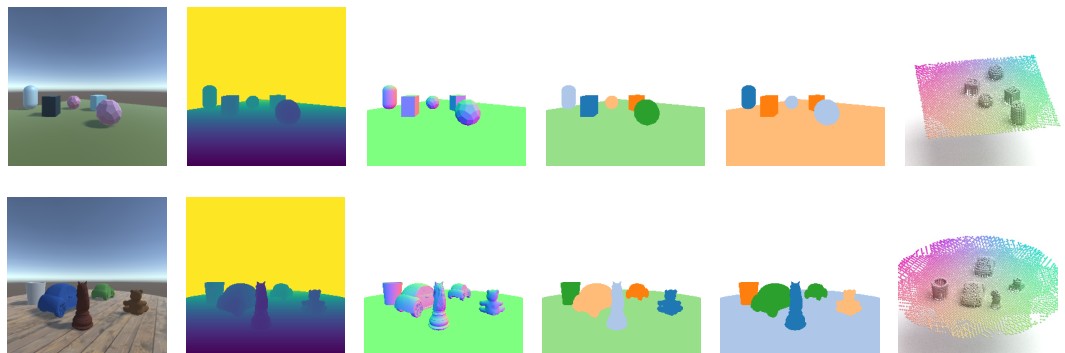

Figure 10: Data captured by each camera in UOR dataset (top) and UOT dataset (bottom). From left to right are RGB, depth, normal, instance label, semantic label and constructed point cloud. Point clouds are obtained by merging multi-view depth images. Instance labels and semantic labels are used to train PointGroup (Jiang et al., 2020) baseline. RGB images and normal maps are not used in this work.

To show that our dataset is indeed challenging for models capturing spatial structure correlations, we plot the empirical **closest neighbor distance distribution** of our data generation process. To obtain the distribution, for each object in each scene, we note down the distance between the **center** of this object and the **center** of its closest neighbor (the surface to surface distance is hard to compute). Thus, distance zero means a complete overlapping between two objects, which is not physically plausible. In our UOR and UOT dataset the minimum distance is set to one (not applicable to Object Matrix layout). With object dimensions specified in Table 13 and Table 14, we consider distance below 2 to be extremely close. Thus, reading from Fig.11, for scenes containing 2-5 objects, 25 percent of objects are spawned close to at least one other object. For scenes containing 6-12 objects, the number goes up to more than 60 percent.

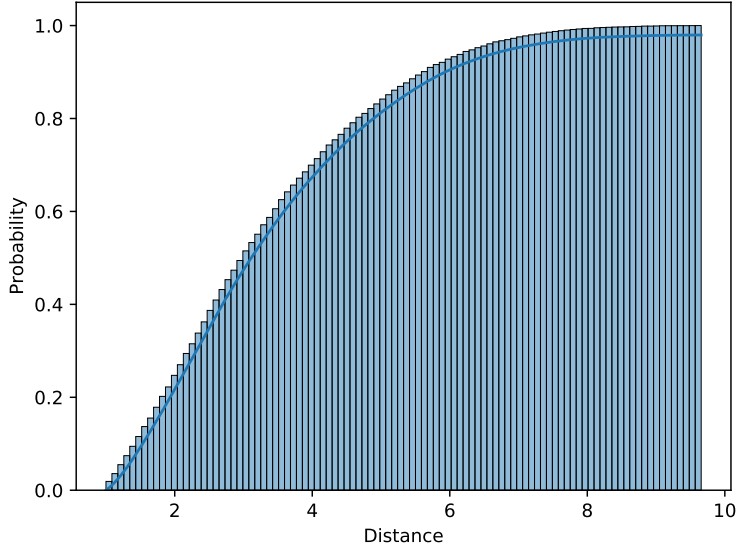

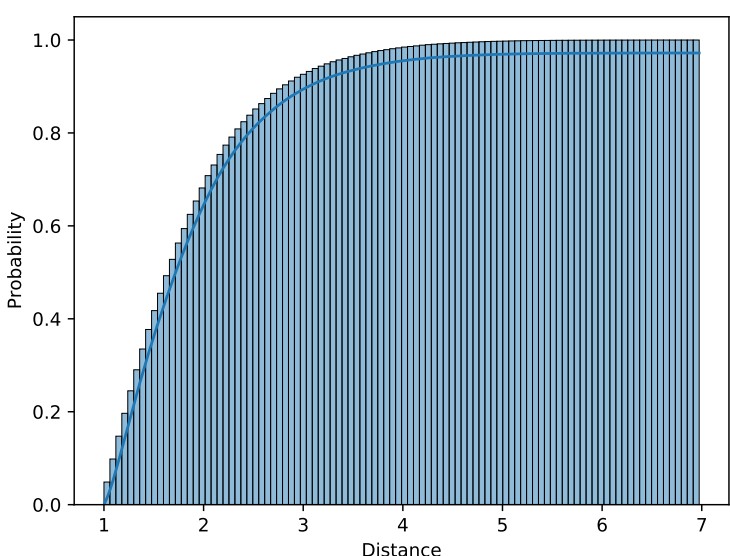

Figure 11: Closest neighbor distance distribution of our data generation process for 2-5 objects (above), and for 6-12 objects (below)

Table 13: UOR object pool.

| geometry | figure | min length/max length | min width/max width | min height/max height |
|---|---|---|---|---|
| Capsule (standing) |  | 0.75/1.25 | 0.75/1.25 | 1.5/**2.5** |
| Capsule (flat) |  | 1.5/**2.5** | 1.5/**2.5** | 0.75/1.25 |
| Cube |  | 0.75/1.25 | 0.75/1.25 | 0.75/1.25 |
| Cylinder (standing) |  | 0.75/1.25 | 0.75/1.25 | 1.5/**2.5** |
| Cylinder (flat) |  | 1.5/**2.5** | 1.5/**2.5** | 0.75/1.25 |
| Hexagonal Prism |  | 1/1.33 | 1/1.33 | 0.5/0.83 |
| Sphere |  | 0.75/1.25 | 0.75/1.25 | 0.75/1.25 |
| Rhombicosidodecahedron |  | 0.75/1.25 | 0.75/1.25 | 0.75/1.25 |
| Square Antiprism |  | 1/1.5 | 1/1.5 | 0.67/1 |
| Triangular Prism (standing) |  | 0.75/1.5 | 0.75/1.5 | 0.75/1.5 |
| Triangular Prism (flat) |  | 0.75/1.5 | 0.75/1.5 | 0.75/1.5 |

Table 14: UOT Object pool. Some object meshes are obtained from ai2thor (Kolve et al., 2017) environment.

| object | figure | min length/max length | min width/max width | min height/max height |
|---|---|---|---|---|
| Chess piece |  | 0.69/0.86 | 0.69/0.86 | 1.43/1.80 |
| Bear |  | 1.1/1.46 | 0.71/0.95 | 1.06/1.41 |
| Box |  | 0.98/1.23 | 0.98/1.23 | 0.67/0.84 |
| Car |  | 1.17/**1.96** | 0.74/1.24 | 0.70/0.78 |
| Cup |  | 0.87/1.10 | 0.70/0.87 | 0.92/1.16 |
| Kettle |  | 0.85/1.13 | 0.69/0.91 | 1.01/1.35 |
| Pot |  | 1.06/1.42 | 0.88/1.17 | 0.80/1.07 |
| Tissue |  | 1.05/1.31 | 0.95/1.18 | 1.16/1.44 |
| Toaster |  | 0.58/0.78 | 1.01/1.36 | 0.61/0.81 |

# E    MORE RESULTS

In this section, we show more segmentation results of our model. To demonstrate that our model can handle non-trivial scene layout, we apply SPAIR3D on classic Object Room dataset (Eslami et al., 2018) and show quantitative results in Fig. 12.

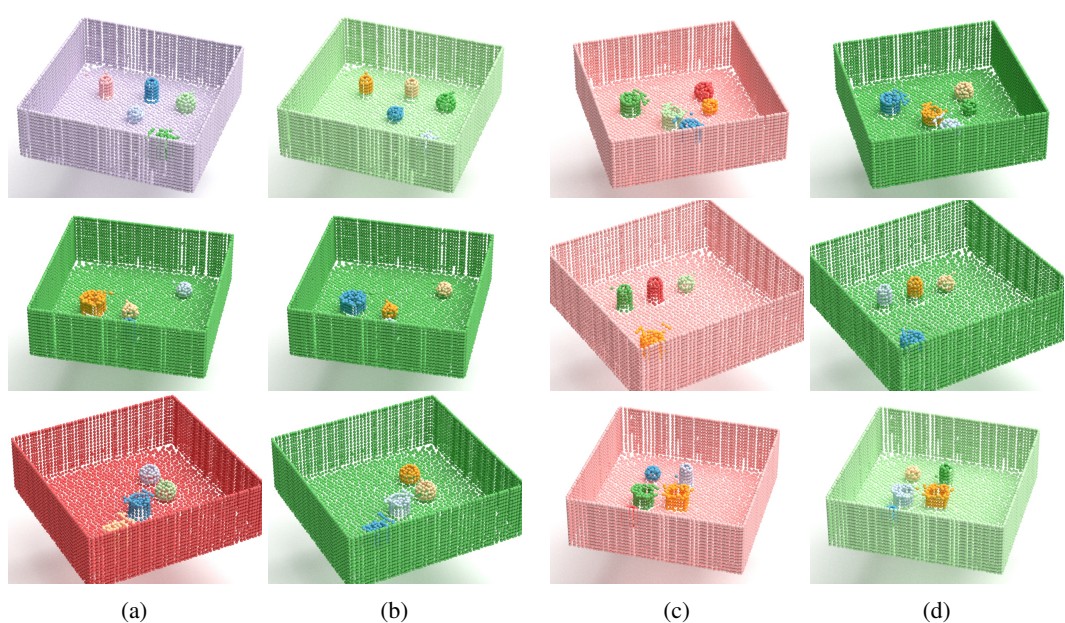

|     (a)     |     (b)     |     (c)     |     (d)     |

Figure 12: Scene layout in Object Room dataset has four walls. SPAIR3D groups four walls together with the floor in each scene as the scene layout component. Column (a) and column (c) are instance labels. Column (b) and column (d) are the corresponding SPAIR3D segmentation.

We show more test cases where SPAIR3D achieves above 0.8 SC scores on UOR in Fig. 13 and UOT in Fig. 14. To provide a more comprehensive inspection, we further display segmentation results on scenes where SPAIR3D achieves below 0.7 SC scores on UOR in Fig. 15, and UOT in Fig. 16, respectively. Segmentation results demonstrated below, when examined together with performance distribution shown in Fig. 4, confirm that our model achieves high quality segmentation on majority scenes.

More segmentation results on scenes with $6 - 12$ objects are shown in Fig. 17 and Fig. 18. In Fig. 19, we show more examples on *Object Matrix* scenes to further demonstrate the scalability of our model.

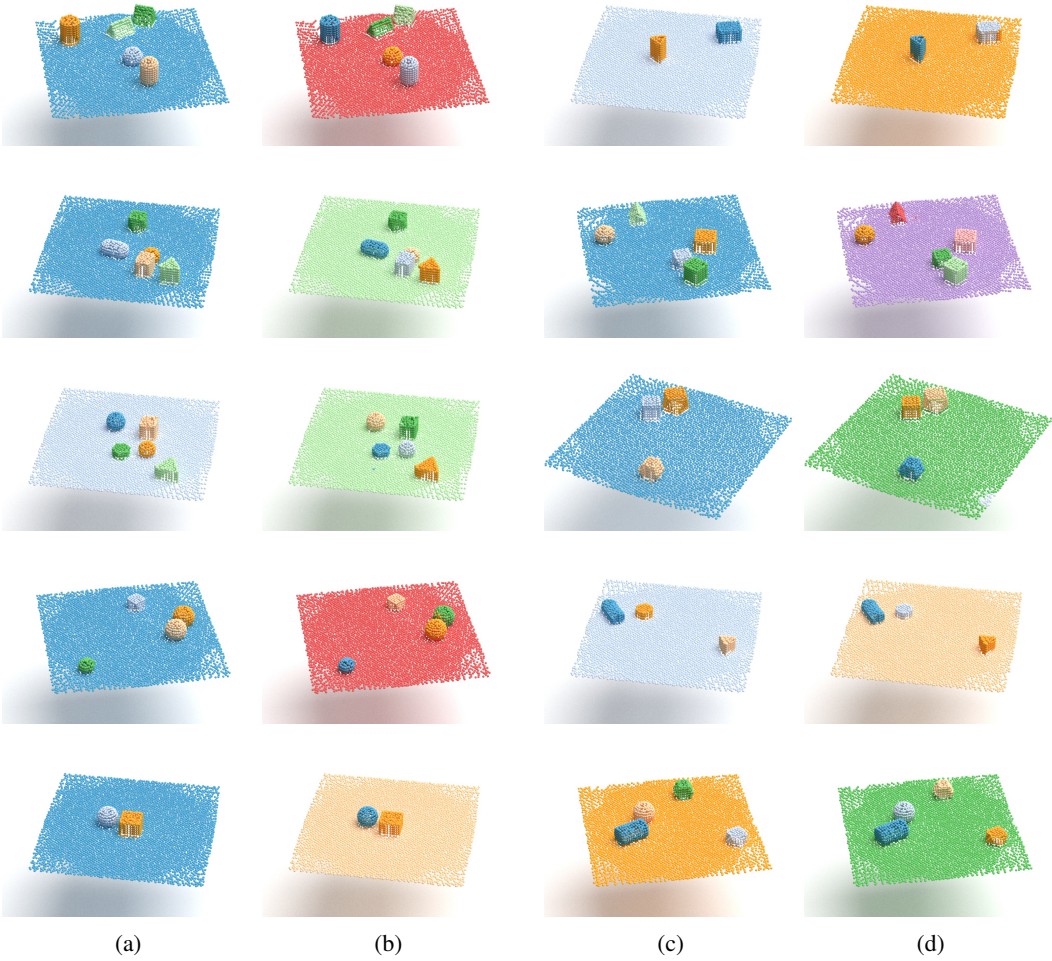

(a)        (b)        (c)        (d)

Figure 13: Typical UOR test cases that achieves above 0.8 SC. Column (a) and column (c) are instance labels. Column (b) and column (d) are the corresponding SPAIR3D segmentation.

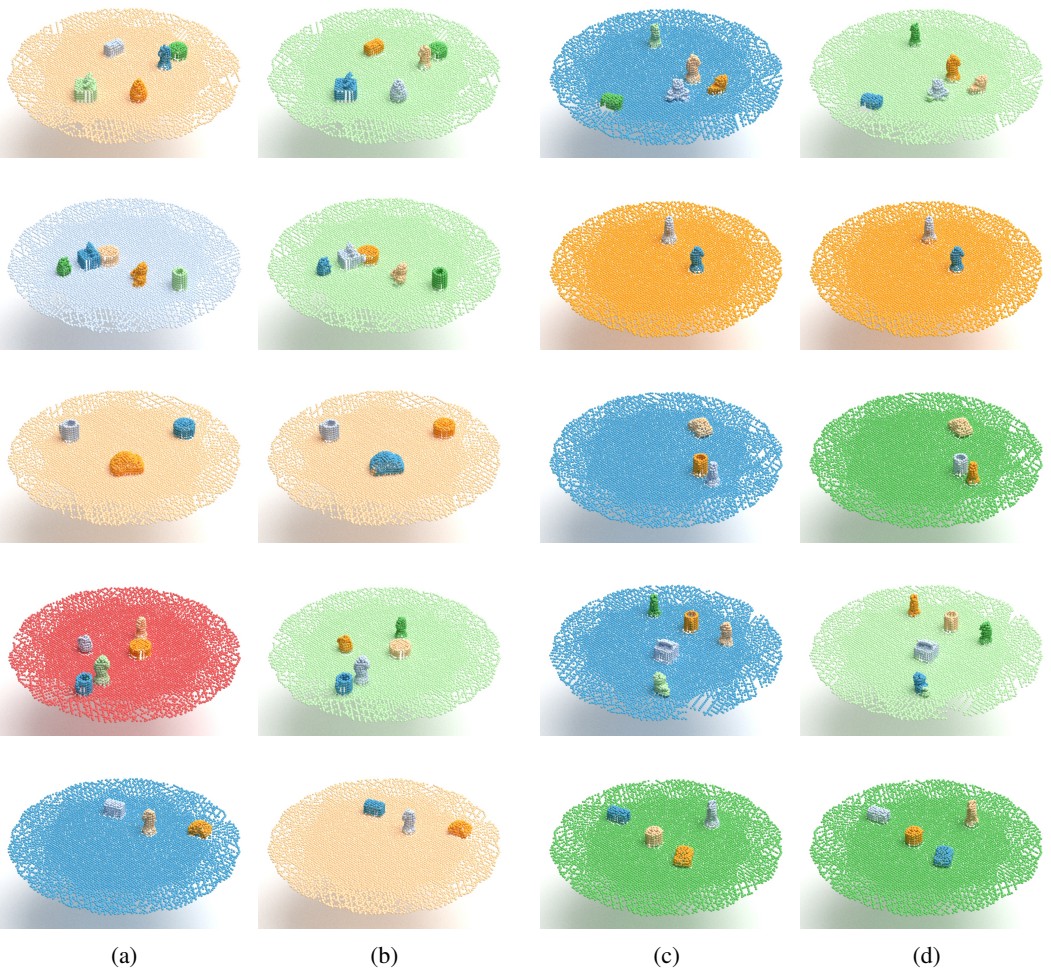

(a)                    (b)                    (c)                    (d)

Figure 14: Typical UOT test cases that achieve above 0.8 SC. Column (a) and column (c) are instance labels. Column (b) and column (d) are the corresponding SPAIR3D segmentation results.

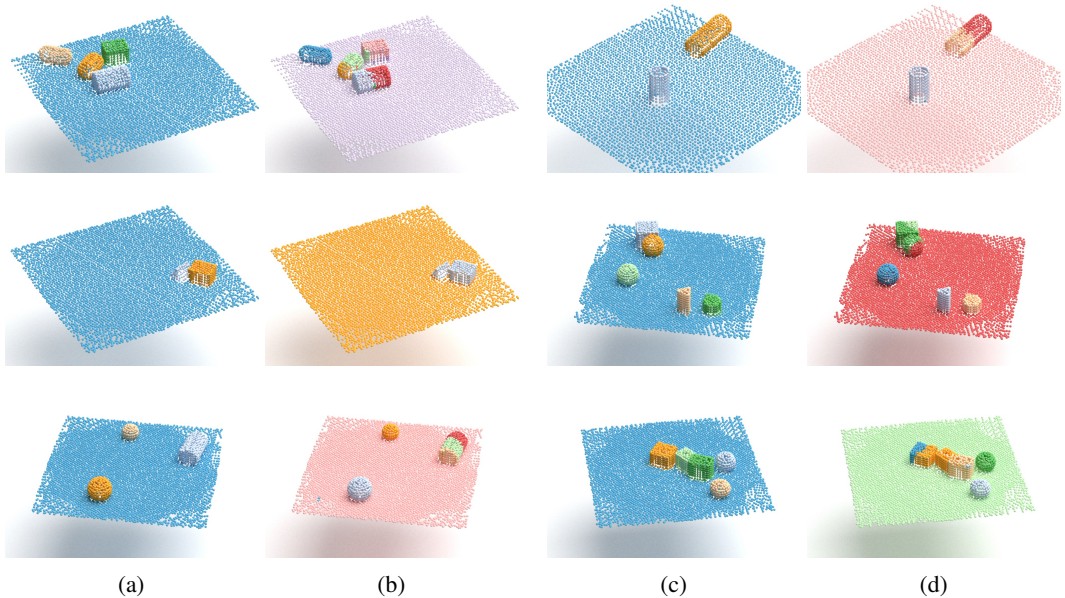

(a)         (b)         (c)         (d)

Figure 15: UOR test cases that achieves below 0.7 SC. Column (a) and column (c) are instance labels. Column (b) and column (d) are the corresponding SPAIR3D segmentation. Failed cases reflect that (1) objects clustered together are more vulnerable to mis-segmentation due to complicated local spatial structure layout (2) and objects with extreme dimensions, e.g. cylinder, are vulnerable to over-segmentations. Note that currently all derivatives of SPAIR (Crawford & Pineau, 2019) are sensitive to object size.

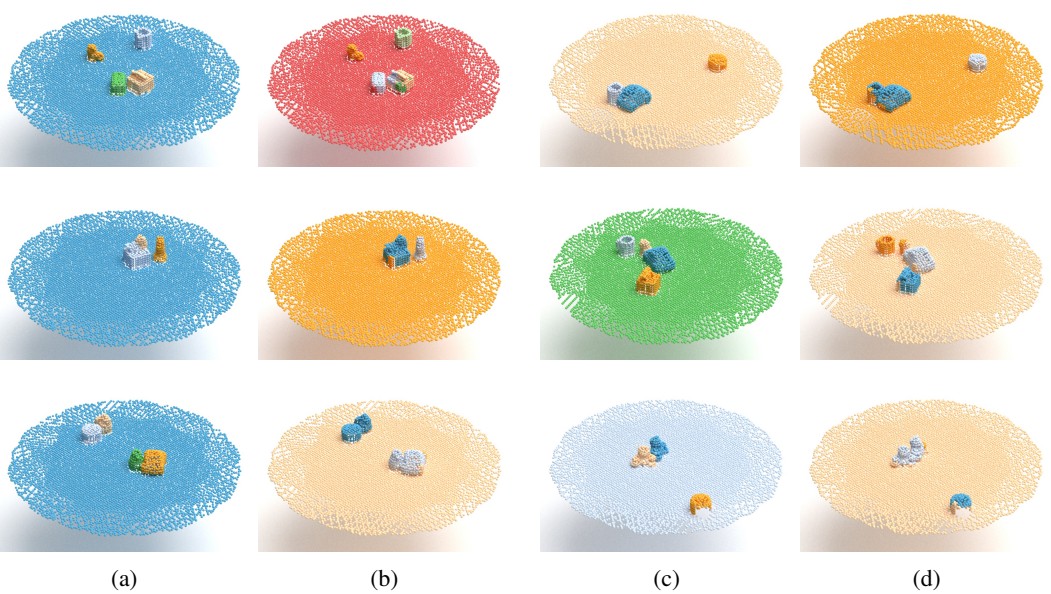

(a)         (b)         (c)         (d)

Figure 16: UOT test cases that achieve below 0.7 SC. Column (a) and column (c) are instance labels. Column (b) and column (d) are the corresponding SPAIR3D segmentation results.

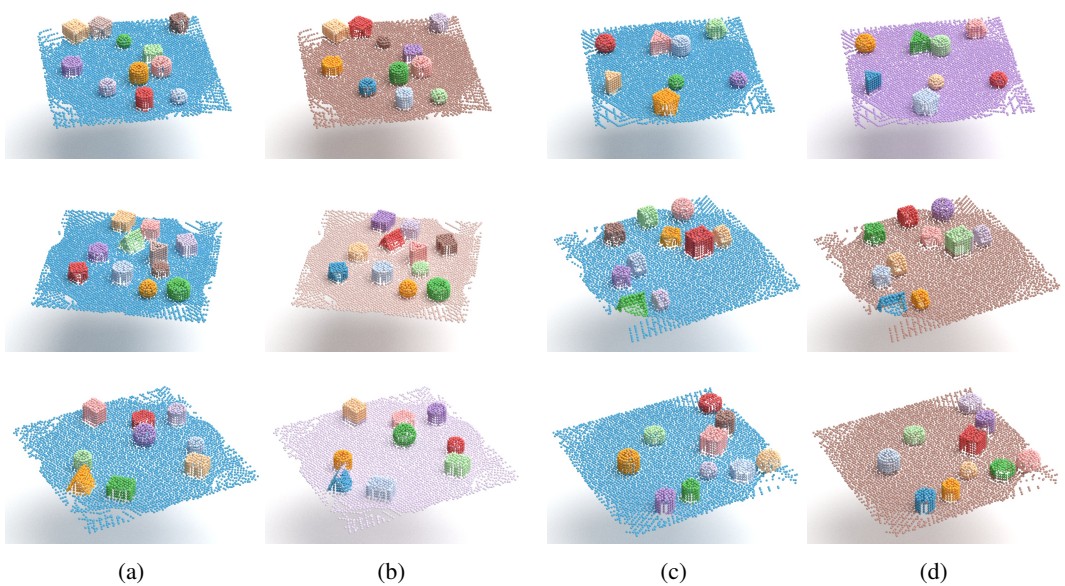

(a)  (b)  (c)  (d)

Figure 17: More results on UOR scenes with 6-12 objects. Column (a) and column (c) are instance labels. Column (b) and column (d) are the corresponding SPAIR3D segmentation.

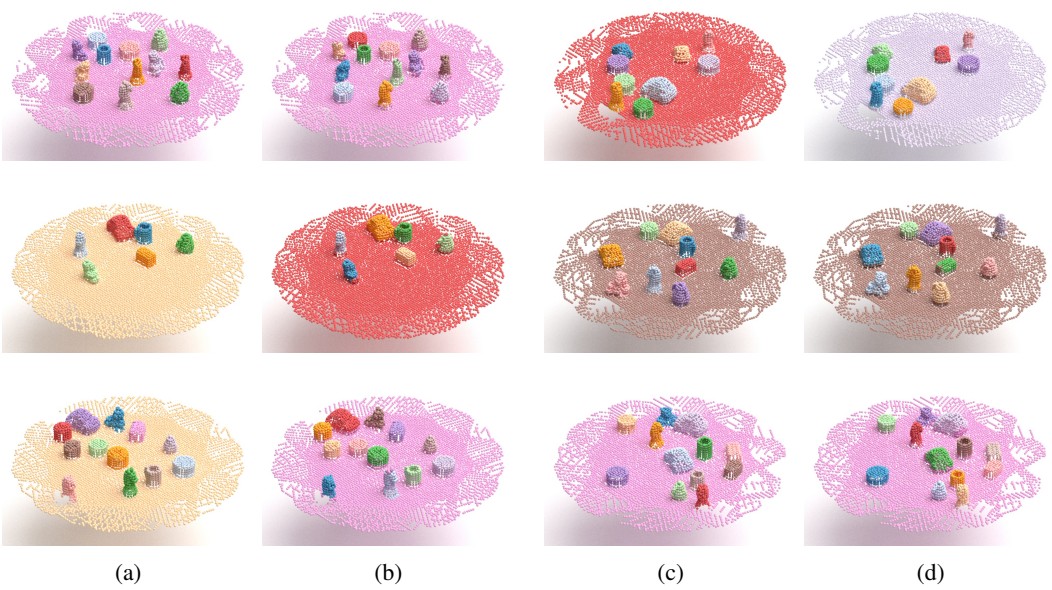

(a)  (b)  (c)  (d)

Figure 18: More results on UOT scenes with 6-12 objects. Column (a) and column (c) are instance labels. Column (b) and column (d) are the corresponding SPAIR3D segmentation.

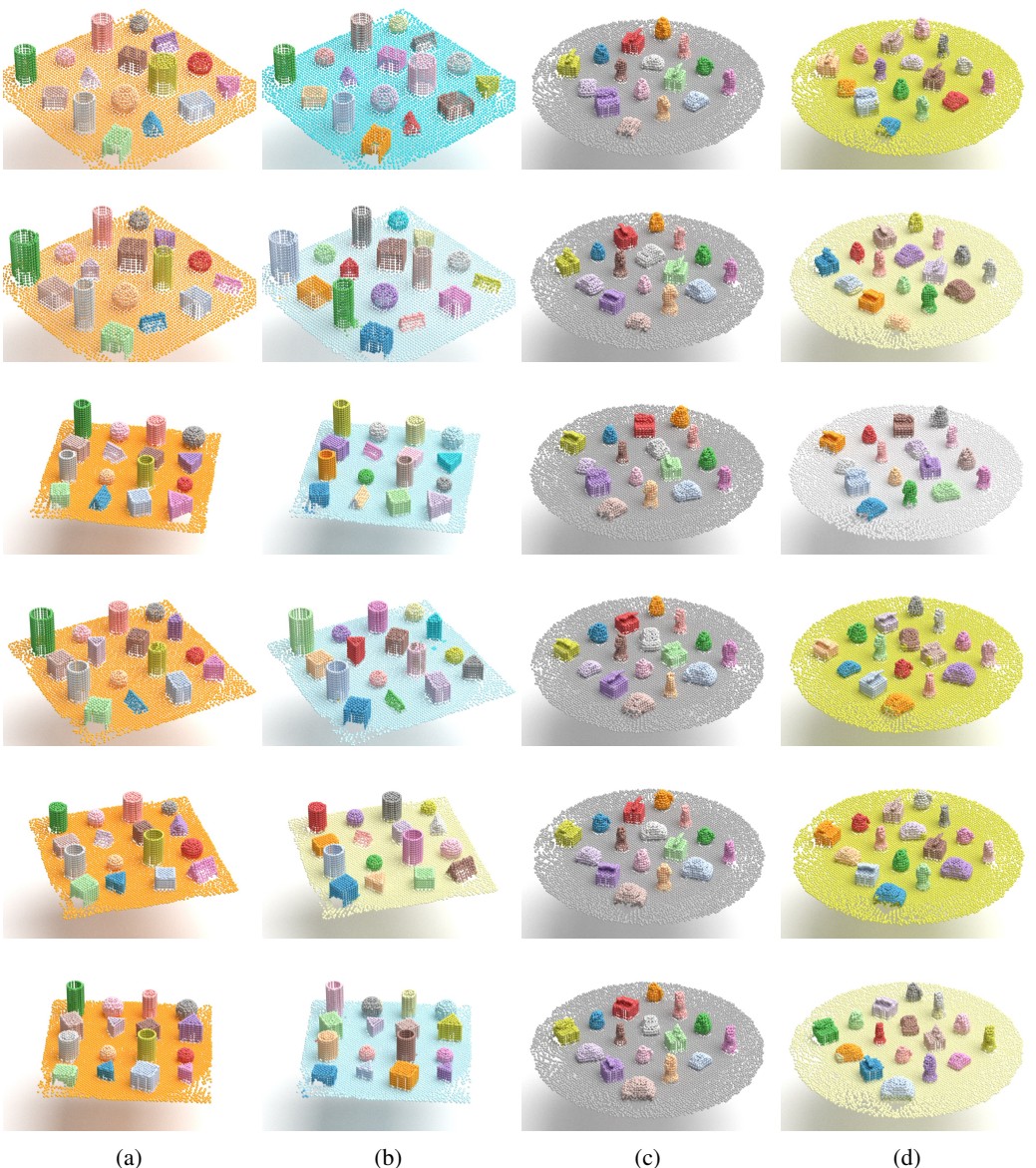

| (a) | (b) | (c) | (d) |

Figure 19: More results on UOR and UOT Object Matrix scenes. Column (a) and column (c) are instance labels. Column (b) and column (d) are the corresponding SPAIR3D segmentation results.

