# OpenReview forum: "Spatially Invariant Unsupervised 3D Object-Centric Learning and Scene Decomposition"
_ICLR.cc/2022/Conference — ICLR 2022 Submitted_

### Official Review · Reviewer_oa19 · 2021-10-26

**Correctness:** 3
**Technical Novelty And Significance:** 2
**Empirical Novelty And Significance:** 2
**Recommendation:** 5
**Confidence:** 4

**Main Review:**

Strengths：
1. This paper proposes a novel Chamfer Mixture loss formulated in a probabilistic framework.
2. The proposed method achieves astonishing performance comparable to a supervised baseline on two simple datasets.

Weakness：
1.	This paper is built on the SPAIR framework and focuses on point cloud data, which is somehow incremental.
2.	There is no ablation study to validate the effectiveness of the proposed components and the loss.
3.	It is hard to follow Sec. 3.2. The author may improve it and give more illustrations and examples.
4.	It is unclear how the method can work and decompose a scene into different objects. I did not see how Chamfer Mixture loss can achieve this goal. More explanation should go here.


**Summary Of The Paper:**

This paper proposes a VAE-based unsupervised generative model, to achieve 3D object-centric learning as well as 3D scene decomposition. The method divides a 3D scene into voxel grids and generates latent features to reconstruct an original point cloud. A chamfer Mixture loss is proposed to train the model.

**Summary Of The Review:**

The idea to decompose a scene in an unsupervised way is novel. However, the proposed method is mainly based on SPAIR and the only technical contribution seems to be Chamfer Mixture loss. It is unclear what is the advantage of this loss design over traditional chamfer distance and how it can help scene decomposition. Moreover, I did not see further analysis on different proposed components of the method and inspiring conclusion for the community.

---

> ### Author Response · Authors · 2021-11-18
> **Reply to reviewer oa19**
>
> We thank the reviewer for their time.
>
> *1. While our work is indeed built on SPAIR, at the beginning of Sec 3, we detail the gap between this work and previous literature, the major challenges, and our novel solutions to those challenges.
>
> *2. The reviewer may miss Sec 4.5 and 4.6.
> The effectiveness of the multi-layer PointGNN z_pres decoder is proven by the ablation study in Sec, 4.5.
> The effectiveness of Point Graph Decider is demonstrated in Sec, 4.6.
> The main difference between our Chamfer mixture loss and Chamfer distance is at the backward exponential weighting.
> The proposed glimpses inevitably contain points that do not belong to the corresponding objects.
> The backward exponential weighting makes sure that such points have no influence on the likelihood and do not pull the generated points toward their location.
> In single object reconstruction tasks, this is not an issue since all points are known to belong to the one object in the scene.
> On the UOR dataset, we rerun our experiments without exponential weighting, and the ARI score drops from 0.915 to 0.552.
> To further evaluate our model, we run additional experiments.
> The results are shown in the newly uploaded supplementary zip file.
>
> *3. The complete definition of ELBO is provided in the last paragraph of Sec. 3.3.
> Due to limited space, we give the definition of the L_KL term in appendix A equation (2).
> The derivation of ELBO follows the vanilla variational inference framework.
> We will make the section clear.
>
> *4 Similar to previous works on generative model-based scene decomposition [1][[2][3], our model relies on the concept of "information bottleneck". Unsupervised object-centric learning is essentially an attempt to define objectness. While rule-based unsupervised segmentation algorithms code low-level heuristics/human prior knowledge directly, generative model-based object-centric models are built on a high-level heuristic. That is, the concept of "objectness" is a consequence of agents trying to model the world as accurately as possible but also trying to process as little information as possible due to the limited computational power.
>
> That is, the term "object" should refer to matters that are highly correlated with each other. Correlation means both predictability and compressibility. Using VAE is a good way to enforce such high-level heuristics. In fact, a VAE effectively compresses the input and tries to reconstruct it under an information budget. Thus, highly correlated parts of inputs are forced to be compressed together, and the union of those parts is considered an object. See beta-VAE related work [4][5] for the in-depth introduction.
>
> From the technical perspective, the segmentation performance is a consequence of the balance between multiple terms.
>
> (1) reconstruction likelihood: One trivial suboptimal of the forward likelihood is to distribute reconstruction points densely and evenly across the entire space so that for any input point there exists a close reconstruction point. One trivial suboptimal of the backward loss is to distribute all reconstruction points close to one input point. The balance between the number of input points and reconstruction points suggests the trivial suboptimal for one loss is an extremely bad solution for the other loss. Also, note that the reconstruction likelihood as a whole encourages over-segmentation. More glimpses lead to smaller parts that are easier to reconstruct compared with the whole object with a non-trivial structure.
>
> (2) KL term of z_pres: to counter the over-segmentation tendency, the prior of z_pres encourages glimpse rejection. Note that once the glimpse is rejected, this glimpse will not increase the final loss.
>
> (3) KL term of z_what: The KL term of z_what implements the information bottleneck idea. While the KL term of z_pres encourages glimpse rejection, the bottleneck makes sure that one glimpse does not have enough capacity to model all objects well. Most importantly, z_what of scene layout branch makes sure that scene glimpse along cannot achieve a high-quality reconstruction of the entire scene. Compared with AE, VAE allows a trade-off between reconstruction quality and information budget in a dynamic way through the training.
>
> (4) KL term of z_where: This term serves as a tie-breaker. When multiple glimpses try to model the same object, this KL term encourages the closest glimpse to take the object.
>
> (5) KL term of glimpse size and the boundary loss: The size of glimpses is encouraged to shrink so that each object only contains points that absolutely belong to this object. To be more specific, if one part of an object (or floor) is already modeled by one glimpse well, then excluding this part from other glimpses will not decrease the likelihood. The boundary loss changes the hard "excluding" process into its soft version. Thus, a point being gradually excluded will change the mask in a differentiable way.
>
> A balance between them gives a good segmentation result.

---

> > ### Author Response · Authors · 2021-11-19
> > **Reply to reviewer oa19 continue**
> >
> > We believe that our work is inspiring for the community for two reasons.
> >
> > (1) For the 3D point cloud community, we show that it is possible to learn a generative model that can naturally decompose scenes.
> > Almost all previous (single object) point cloud generation pipeline requires the reconstruction target to be centered and normalized.
> > When it comes to object-centric scene generation, it is almost impossible to require all parts of the raw input to be centered and normalized at the same time.
> > The local attention mechanism and the spatial invariant property of our model make sure that all objects are reconstructed in their local object coordinate systems instead of the global coordinate.
> >
> > (2) For the object-centric learning community, we show that the generative model-based/the VAE based object-centric learning idea can be applied to not only color but to spatial structure as well.
> > With more and more data modalities added, the object-centric representation should be more and more meaningful.
> >
> > Hopefully, the majority of your concerns are addressed.
> > Please let us know if you have further concerns.
> >
> > [1] Klaus Greff, Rapha ̈el Lopez Kaufman, Rishabh Kabra, Nick Watters, Christopher Burgess, Daniel Zoran, Lo ̈ıc Matthey, Matthew M Botvinick, and Alexander Lerchner Multi-object representation learning with iterative variational inference. In ICML, 2019.
> >
> > [2] Paul Henderson and Christoph H. Lampert. Unsupervised object-centric video generation and decomposition in 3D. In NeurIPS, 2020.
> >
> > [3] Zhixuan Lin, Yi-Fu Wu, Skand Vishwanath Peri, Weihao Sun, Gautam Singh, Fei Deng, Jindong Jiang, and Sungjin Ahn. Space: Unsupervised object-oriented scene representation via spatial attention and decomposition. In ICLR, 2020.
> >
> > [4] rina Higgins, Lo ̈ıc Matthey, Arka Pal, Christopher Burgess, Xavier Glorot, Matthew M Botvinick, Shakir Mohamed, and Alexander Lerchner. beta-vae: Learning basic visual concepts with a constrained variational framework. In ICLR, 2017
> >
> > [5] Christopher P. Burgess, Irina Higgins, Arka Pal, Lo ̈ıc Matthey, Nick Watters, Guillaume Desjardins, and Alexander Lerchner. Understanding disentangling in beta-vae. ArXiv, abs/1804.03599, 2018.

---

### Official Review · Reviewer_orM7 · 2021-10-31

**Correctness:** 4
**Technical Novelty And Significance:** 3
**Empirical Novelty And Significance:** 3
**Recommendation:** 6
**Confidence:** 4

**Main Review:**

# Strengths
- The overall writing is clear.
- The paper is the first to extend 2D object-centric representation learning to 3D point clouds. The Chamfer Mixture Loss is a reasonable adaptation for variational training.

# Weaknesses/Questions
1. Lack of baselines. Although there are no direct 3D object-centric representation learning baselines, 3D object discovery is not an unexplored field. Apart from "Unsupervised Discovery of Repetitive Objects", a common baseline in 3D object detection is to cluster/group 3D points according to their spatial relationship. Concretely, the authors can compare with a baseline, where the background/scene layout (planes in the paper's datasets) is first detected by a RANSAC and the rest objects are separated by DBSCAN/MeanShift (hyperparameters need to be tuned/searched). Although such a baseline is not scalable to more complicated scenes, SPAIR3D is not shown to have stronger scalability currently. Besides, the authors can introduce some 3D over-segementation baselines (e.g., [1], [2])

2. How important is VAE in the object-centric representation learning framework? It might be a general question (also for 2D counterparts). From some experience, I suspect that the prior of model structure, like sliding window in 2D/3D grid and relative center prediction (techniques studied in 2D/3D detection), is the most important. VAE, especially the probabilistic model, mainly provides a sampling procedure during training, which might be considered as a search process (Monto-Carlo sampling). An easy way to check is to remove KL divergence terms between the prior and posterior probability of latent codes, and only maximize posterior probabilities of observations.

3. I am a little confused about the claim "Unfortunately, the Chamfer distance does not fit into the variational-inference framework". There is no detailed explanation about it. From my understanding, the Chamfer Mixture Loss just introduces additional variance term and changes the distance to probability, since the variance term is a hyperparameter rather than per-point prediction and the likelihood is only based on the best/maximum matched point.

4. As mentioned in Sec 4.1, the point cloud is fused from 10 different views. Can SPAIR3D work on partial 3D scenes? Theoretically, current SPAIR3D may have difficulty since the Chamfer Mixture Loss ignores visibility and it becomes much harder to recognize repetitive objects.

# References
1. Toward better boundary preserved supervoxel segmentation for 3D point clouds
2. Superpoint Network for Point Cloud Oversegmentation

**Summary Of The Paper:**

The paper tackles unsupervised 3D scene decomposition/object discovery from point clouds. It proposes SPAIR3D, inspired by the 2D counterpart SPAIR, to factorize a 3D point cloud into a spatial object-centric mixture model. It derives the Chamfer Mixture Loss to fit 3D reconstruction into the variational training pipeline. It introduces two customized point-cloud datasets, and compares with a supervised baseline PointGroup.

**Summary Of The Review:**

To the best of my knowledge, the paper is the first to tackle 3D object-centric scene decomposition (unsupervised object discovery). It is well written and reasonably extends SPAIR to 3D scenes with new challenges only in 3D cases. However, the baselines are not carefully chosen, and the scalability of the proposed SPAIR3D is questionable. Overall, I think the paper is slightly above the acceptance threshold.

---

> ### Author Response · Authors · 2021-11-18
> **Reply to reviewer orM7**
>
> We thank the reviewer for the feedback.
> Below we address the concerns.
>
> 1. Rule-based baseline:
> Yes, for the UOR and UOT datasets, the rule-based segmentation method should indeed achieve good segmentation results.
> However, we consider our work an attempt to achieve object-centric learning on point cloud data.
> The object representation embedding is shown in figure 7 also shows that our learned representation is meaningful.
> The rule-based baseline does not learn an object-centric representation that can be used for downstream tasks.
> Segmentation is only an application of our method and is evidence that our model indeed learns representation in an object-centric way.
> Also, the rule-based baselines do not allow the sampling of new objects.
> Scalability: our foreground VAE model is agnostic to the absolute coordinate of input points and only works on a local receptive field.
> Thus, it is spatially invariant, just like a CNN layer.
> A CNN layer can take a 2D image with an arbitrary size, so can our foreground VAE.
> Our model essentially learns a probability distribution p(o|z) where o is the observation and z is the latent variable.
> This probability interpretation makes our model compatible with other probabilistic frameworks like SLAM.
> We thank the reviewer for bringing the related over-segmentation baselines to our attention.
>
> 2. VAE is crucial to representation learning, and this result is established by well-cited papers [1][2].
> We detailed the important role of VAE and KL terms in point 6 of our replies to reviewer yyA6.
>
> 3. The main difference between our Chamfer mixture loss and Chamfer distance is at the backward exponential weighting.
> The proposed glimpses inevitably contain points that do not belong to the corresponding objects.
> The backward exponential weighting makes sure that such points have no influence on the final likelihood and thus do not pull generated points toward their location.
> As a consequence, those points do not have effects on glimpse center and size prediction.
> In single object reconstruction tasks, this is never an issue since all points are known to belong to the one object in the scene.
> On the UOR dataset, we rerun our experiments without exponential weighting, and the ARI score drops from 0.915 to 0.552.
>
> 4. Partial observation:
> Your conclusion about partial observation is correct.
> Even with a complete supervision point cloud, partial observation is quite challenging for our model for two reasons.
> (1) With partial information it is hard to determine the correct object location.
> (2) With partial information it is hard to determine the number of points to generate for each glimpse.
> Thus, our current model cannot handle partial observations.
> To further evaluate our model, we run additional experiments.
> The results are shown in the newly updated supplementary zip file.
>
> Hopefully, the majority of your concerns are addressed.
> Please let us know if you have further concerns.
>
> [1] rina Higgins, Lo ̈ıc Matthey, Arka Pal, Christopher Burgess, Xavier Glorot, Matthew M Botvinick, Shakir Mohamed, and Alexander Lerchner. beta-vae: Learning basic visual concepts with a constrained variational framework. In ICLR, 2017
>
> [2] Christopher P. Burgess, Irina Higgins, Arka Pal, Lo ̈ıc Matthey, Nick Watters, Guillaume Desjardins, and Alexander Lerchner. Understanding disentangling in beta-vae. ArXiv, abs/1804.03599, 2018.

---

> > ### Comment · Reviewer_orM7 · 2021-11-23
> > **Reply to the authors' rebuttal**
> >
> > Thanks for the authors' reply.
> >
> > 1. The reason why it is better to compare with a heuristic-based method is that it is really a strong and in fact generalizable baseline on 3D scenes. A very special property of 3D scenes is that objects are usually spatially separable. For example, such cluster-based segmentation (DB-Scan, Meanshift, connected component) methods can work quite well even on both indoor and outdoor real scenes (like ScanNet and SemanticKitti). With slight efforts on parameter tunning, such baselines can perform on par with learning-based methods (e.g. GSPN) on ScanNet instance segmentation. Moreover, the proposed SPAIR3D in fact shares some underlying ideas with GSPN[1], except GSPN is a supervised method.
> > From my perspective, the heuristic-based method is usually spatial-invariant as well, and can deal with input of arbitrary size. Since the SPAIR3D also needs hyperparameters for glimpse/cell size, it is hard to tell whether SPAIR3D is a more robust and generalizable method.
> > However, object-centric representation is indeed useful, which can distinguish SPAIR3D from those spatial heuristic-based methods. Thus, I agree with the contribution of SPAIR3D, but can not agree with the explanation why the authors do not compare with those strong baselines.
> >
> > 2. I think beta-VAE focuses on the disentanglement of latent factors; however, in this work, such disentanglement is implicitly achieved by the inductive bias of network architectures. Thus, it is quite different. Please correct me if I am wrong.
> >
> > [1] Gspn: Generative shape proposal network for 3d instance segmentation in point cloud

---

> > > ### Author Response · Authors · 2021-11-24
> > > **Reply to reviewer orM7**
> > >
> > > We thank reviewer or7M for their feedback
> > >
> > > 1. Following your suggestion, we introduce a rule-based baseline on the cleaned S3DIS dataset (see the supplementary zip file for details) with floors manually removed.
> > > We tried both the Meanshift and the DBSCAN algorithm.
> > > The best results are achieved by the Meanshift algorithm after doing the grid-search on the bandwidth parameter for chairs and tables separately.
> > > With the bandwidth tuned for chair (0.05), the best mIoU is Chair:0.634, Table:0.297, Sofa:0.272.
> > > With the bandwidth tuned for table (0.1), the best mIoU is Chair:0.535, Table:0.442, Sofa:0.471.
> > > As stated in the supplementary zip file, our model achieves Chair: 0.603, Table:0.375, and Sofa: 0.482.
> > > Note that the bandwidth parameter of the Meanshift algorithm reflects the average object size.
> > > Thus, when the bandwidth is tuned for chairs whose sizes are much smaller on average compared to tables, tables and sofas are largely over-segmented, hence resulting in the low value in mIoU.
> > > When the bandwidth is tuned for tables, chairs tend to be under-segmented.
> > > For our work, while the average sizes of tables and sofas are much larger than the maximum glimpses sizes, SPAIR3D still tries to expand the glimpses sizes to better model larger objects while keeping the total number of glimpses low.
> > > We believe that the relaxation of the glimpses sizes limitation is a key research direction for our future work.
> > >
> > > 2. Yes, beta-VAE is known for better feature disentanglement, and in our work, the major disentanglement is enforced by our network structure.
> > > However, due to the KL regularisation,  VAE, in general, presents a more structured latent space where each class tends to occupy one connected region, as shown in Fig. 3 in our paper.
> > > VAE is adopted by not only derivations of SPAIR but almost all generative model based unsupervised object-centric representation learning frameworks [1,2,3,4,5,6].
> > > It is fair to say that VAE is the default choice in the object-centric representation learning community.
> > > The KL term also allows a dynamic trade-off between information budget and reconstruction likelihood during the training process either via weights annealing (this work) or in a systematic way [7].
> > > Such fine level control is not possible with AE where one can only set the dimension of the latent variable.
> > > In this work, we want to demonstrate that the SPAIR framework (with VAE being a core component) can be extended to point cloud data which is confirmed by our experimental results.
> > > Potential modification/improvement on the SPAIR framework, in general, is beyond the scope of our work.
> > >
> > > [1] Christopher Burgess, Loic Matthey, Nicholas Watters, Rishabh Kabra, Irina Higgins, Matt Botvinick,
> > > and Alexander Lerchner. Monet: Unsupervised scene decomposition and representation. ArXiv,
> > > abs/1901.11390, 01 2019. URL https://arxiv.org/abs/1901.11390.
> > >
> > > [2] Eric Crawford and Joelle Pineau. Spatially invariant unsupervised object detection with convolutional
> > > neural networks. AAAI, 33:3412–3420, 07 2019. doi: 10.1609/aaai.v33i01.33013412.
> > >
> > > [3] Eric Crawford and Joelle Pineau. Exploiting spatial invariance for scalable unsupervised object
> > > tracking. AAAI, 34:3684–3692, 04 2020. doi: 10.1609/aaai.v34i04.5777.
> > >
> > > [4] Martin Engelcke, Adam R. Kosiorek, Oiwi Parker Jones, and Ingmar Posner. Genesis: Generative
> > > scene inference and sampling with object-centric latent representations. In ICLR, 2020. URL
> > > https://openreview.net/forum?id=BkxfaTVFwH.
> > >
> > > [5] Paul Henderson and Christoph H. Lampert. Unsupervised object-centric video generation and
> > > decomposition in 3D. In NeurIPS, 2020.
> > >
> > > [6] Zhixuan Lin, Yi-Fu Wu, Skand Vishwanath Peri, Weihao Sun, Gautam Singh, Fei Deng, Jindong
> > > Jiang, and Sungjin Ahn. Space: Unsupervised object-oriented scene representation via spatial
> > > attention and decomposition. In ICLR, 2020. URL https://openreview.net/forum?
> > > id=rkl03ySYDH.
> > >
> > > [7] Danilo Jimenez Rezende and Fabio Viola. Taming VAEs. arXiv preprint arXiv:1810.00597, 2018.

---

> > > > ### Comment · Reviewer_orM7 · 2021-11-24
> > > > **Thanks for the update**
> > > >
> > > > Thanks for the timely update.
> > > >
> > > > It is nice to see the comparison against classical heuristic methods, and I suggest it could be included in the appendix to provide readers with more insights into the advantage of learning-based unsupervised scene decomposition methods. Besides, I understand that since SPAIR, most related works have not analyzed how their inductive bias affects the performance compared to VAE (especially probability modeling). From my personal experience, sometimes SPAIR-style framework still can work if there are no latent codes for each glimpse (e.g., a single-shot detection/reconstruction framework). However, there are no strong 2D heuristic baselines. Thus, the comparable performance of heuristic baselines somehow indicates the inductive bias is at least as important as VAE for unsupervised scene decomposition, which is valuable to the community.

---

> > > > > ### Author Response · Authors · 2021-11-27
> > > > > **To reviewer or7M**
> > > > >
> > > > > We thank the reviewer for the suggested baseline and we are glad that the reviewer finds that our results and analysis provide meaningful insights.
> > > > > While openreview no longer allows us to update our writing at the moment,  we will certainly include the new results and the analysis in our appendix.

---

> > ### Comment · Reviewer_orM7 · 2021-11-29
> > **Thanks for the authors' rebuttal**
> >
> > Given the authors' rebuttal, I would like to keep my rating. Although the proposed method is currently limited, e.g., relatively easy dataset without occlusion and strong size prior, it shows the potential of learning-based unsupervised 3D instance segmentation methods (compared against classical heuristic methods).

---

### Official Review · Reviewer_tBSM · 2021-11-01

**Correctness:** 3
**Technical Novelty And Significance:** 3
**Empirical Novelty And Significance:** 3
**Recommendation:** 5
**Confidence:** 4

**Details Of Ethics Concerns:**

Evaluating only one dataset (the synthetic dataset rendered by this paper) is not fair enough. The evaluation dataset would better be different from the training dataset. Comparison with only one prior work is also not enough.

**Main Review:**

The paper proposes an unsupervised object-level segmentation framework from point cloud input. The framework is unsupervised, driven by a Chamfer Mixture Loss. Two training datasets, UOR and UOT are rendered by Unity to train and evaluate the proposed method. In experiments, the method is compared with a supervised method PointGroup, where PointGroup outperforms SPAIR3D, but the performances are comparable.
The unsupervised method and the new loss are good and novel, but I have a few confusions below:
1. Firstly the problem setting is a little confusing, since the framework is unsupervised, I thought that the training data would be captured point clouds or those from other various datasets, since GT is not needed. However, the authors still make a lot of efforts to render UOR and UOT. If so, the GT is easy to get, why do we need an unsupervised framework for these synthetic datasets? On the other hand, the synthesized datasets look relatively simple and objects are placed sparsely and at similar sizes. I would like to see results on more challenging data, and I would like to hear the reason why the training is not performed on real point clouds or existing datasets.
2. In a real scene, objects are often stacked. Would SPAIR3D work and how is it work?
3. Typos: title of 3.1 "Local".


**Summary Of The Paper:**

This paper introduces a framework, SPAIR3D, to decompose a 3D point cloud into several objects (point cloud segmentation). The framework is unsupervised, driven by a Chamfer Mixture Loss. Two training datasets, UOR and UOT are rendered by Unity to train and evaluate the proposed method. In experiments, the method is compared with a supervised method PointGroup, where PointGroup outperforms SPAIR3D, but the performances are comparable.

**Summary Of The Review:**

The idea of unsupervised point cloud segmentation is cool since point clouds are hard to annotate, and they are easy to capture or collect. However, the dataset and evaluations are not quite reasonable to me, as described above. Thus I give a borderline reject for now.

---

> ### Author Response · Authors · 2021-11-18
> **Reply to reviewer tBSM**
>
> We thank the reviewer for the feedback.
> Below we address the concerns.
>
> 1. Unsupervised generative model training tends to require a large amount of data.
> Take widely adopted 2D object-centric learning datasets as examples,
> The CLEVR dataset[1] contains 100K images.
> "Object Room" by deepmind[2] contains 1M scenes.
> Our model is trained on 50K scenes.
> Generating data from a simulated environment make sure that we have enough labeled data to use.
> Note that we do need ground truth labels to evaluate the performance and train our supervised baseline.
> The second reason is that our model works on complete point cloud data.
> Also, in some complicated scenes, color usually serves as a strong indication of objectness.
> Our model relies purely on structure information.
> To further evaluate our model, we run additional experiments.
> The results are shown in the newly updated supplementary zip file.
>
> 2. Our complete point clouds are obtained by merging depth maps.
> Thus, when objects are stacked on top of each other, it is not possible to capture touching surfaces resulting in an incomplete point cloud.
> In this case, by structural correlation alone, it is hard to perform the segmentation well.
> But with other data modalities like RGB, there is a better chance to segment the objects with high accuracy.
>
> 3. We thank the reviewer for pointing out the typo.
>
> Hopefully, the majority of your concerns are addressed.
> Please let us know if you have further concerns.
>
> [1] Justin Johnson, Bharath Hariharan, Laurens van der Maaten, Li Fei-Fei, C. Zitnick, and Ross Girshick. Clevr: A diagnostic dataset for compositional language and elementary visual reasoning. pp. 1988–1997, 07 2017. doi: 10.1109/CVPR.2017.215
>
> [2] Rishabh Kabra, Chris Burgess, Loic Matthey, Raphael Lopez Kaufman, Klaus Greff, MalcolmReynolds, and Alexander Lerchner. Multi-object datasets. https://github.com/deepmind/multi-object-datasets/, 2019

---

> > ### Comment · Reviewer_tBSM · 2021-11-30
> > **Additional comments**
> >
> > Thanks for the responses.
> >
> > After reading the authors' responses and other reviews, I am curious that whether Spair3D can deal with scenes with objects of varying sizes. In the paper, the objects in each scene are similar in size. In real life, it is common that some objects are way larger than others. Is it possible that the Chamfer Mixture Loss can work for this kind of data?
> >
> > I do not quite agree with the authors about the first reason for using synthetic data, you can totally train on unlabeled data to show the power of unsupervised training, and test on synthetic datasets. It is fair to compare on a generated evaluation set which is unseen for all methods. I guess the second reason is the major one, it is common that captured point clouds are often incomplete.
> >
> > When two objects are stacked, if they are not able to be separated due to the missing points on touching surfaces. I am curious that why objects and the ground can be separated, as in all the demos in the paper.
> >
> > Furthermore, it would be good to enlarge the legends in Figure 1, which is a little bit hard to read in the current version.

---

> > > ### Author Response · Authors · 2021-11-30
> > > **Reply to reviewer tBSM**
> > >
> > > 1. As pointed out by the reviewer yyA6, improving the performance of SPAIR3D on a real-world dataset can take some extra effort but is definitely possible.
> > > One key design is to use a voxel grid with multiple scales and transform the generative model to its hierarchical version.
> > > Note that SPAIR3D is already a two-layer hierarchical structure (entire scene, each object).
> > > But this idea is not even implemented in the 2D object-centric learning community where the original SPAIR framework is proposed.
> > > We believe that a pipeline built on this idea can perform well and deal with objects of different sizes.
> > > But that will take more than one paper to complete.
> > > Also, we want to highlight the fact that, the SPAIR framework, among all generative based object-centric learning frameworks, is most suitable to point cloud data since the glimpse structure allows point-cloud reconstruction in its local coordinate while other frameworks reconstruct under the global coordinate. Uncentered point cloud-based object reconstruction is itself a really challenging work.
> > >
> > > 2. As suggested by reviewer yyA6, we conduct additional experiments on the S3DIS dataset. The results can be found in the supplementary zip file. As the first work that extends the SPAIR framework to point cloud, we hope through this work to show the potential to borrow ideas from the generative model-based object-centric learning community.
> > > Reviewer yyA6 and orM7 agree that while with some limitations, this work, being the first work on this line, demonstrates the potential of this idea.
> > >
> > > 3. The ground can be segmented from the objects due to our prior glimpse sizes.
> > > The prior encourages glimpse to shrink and to contain only the parts that cannot be modeled well by other components.
> > > Since the ground can be modeled well by our scene layout branch, there is no reason to keep them as part of foreground objects.
> > > For stacked objects, the concerns are different.
> > > For objects with similar sizes, with the missing surfaces, one glimpse may be able to model them well.
> > > If object sizes are too different, then that falls back to the large object concerns.
> > >
> > > 4. We will enlarge the legends as suggested. Thanks for your suggestion

---

### Official Review · Reviewer_PtZ1 · 2021-11-02

**Correctness:** 3
**Technical Novelty And Significance:** 2
**Empirical Novelty And Significance:** 2
**Recommendation:** 5
**Confidence:** 2

**Main Review:**

### Strengths
#### S1 - Important problem
- Unsupervised 3D object learning is an important important problem. This paper studies a specific case of it, which is detecting 3D objects in point clouds.

#### S2 - Carefully designed framework
- The proposed method is carefully designed for handling the irregular structure of point clouds in a VAE framework. Specifically, it adopts a likelihood-based extension of Chamfer distance as a reconstruction loss for point clouds, and incorporates several recent architectures, including pointCNN and pointGNN.

### Weaknesses
#### W1 - Over-complicated method on simplistic datasets
- A major concern with this paper is the simplicity of the evaluation experiments. The datasets used in this paper appear extremely simple, with almost perfect shapes, uncluttered scenes and clean, planar background, which makes task of foreground object detection seem rather trivial, and hence the learning of 3D object representations. Moreover, compared to the previous studies in 2D images, the 3D scenes here intrinsically alleviate challenges like occlusion and illumination effects. In contrast to this seemingly trivial task, the proposed method appears overly complicated.
- Also, the failure examples shown in appendix (Fig. 15 and Fig. 16) suggests the model is still unable to handle scenes where objects are closer.
- I would suggest increasing the diversity of the 3D scenes to verify the effectiveness of the proposed method, eg, using more complicated geometries for the background, varying the sizes and point densities of the objects, introducing noise to the point clouds etc.


**Summary Of The Paper:**

This paper studies the problem of unsupervised object-centric learning in the context of point clouds. Specifically, the goal is to learn a generative model of 3D objects from a collection of untextured point clouds of multi-object scenes, and this is achieved via a VAE-based generative framework.

This task extends the setup of prior work on image-based 2D object-centric learning to 3D. To be able to handle irregular structures of point clouds, a novel Chamfer Mixture Loss is proposed as reconstruction loss, which essentially extends the Chamfer distance with probability mixture modeling. The resulting model seems to work well on simple scenes with small isolated objects on a plane.

**Summary Of The Review:**

Overall, I think the problem is interesting, and the proposed framework is technical sound, but the results presented on the simplistic datasets cannot sufficiently verify the effectiveness of the method.

---

> ### Author Response · Authors · 2021-11-18
> **reply to reviewer PtZ1**
>
> We thank the reviewer for the feedback.
> Below we address the concerns.
>
> 1. When compared with 2D cases, complete 3D point clouds indeed do not have occlusion, but we also lost color as a strong cue of objectness.
> At the beginning of Sec 3, we detailed the gap between this work and previous literature as well as the major challenges.
> We believe that extending SPAIR to 3D point cloud is not a trivial task.
>
> 2. Via failure examples, we want to show the fact that when our model fails, there tend to be objects that are touching each other or over-size.
> Those two cases are indeed challenging.
> But other examples in the appendix, especially in Figures 17, 18, show that our model can handle touching objects.
> The fact that the performance does not drop when evaluated on scenes with 6-12 objects (with much more touching surfaces) shows that our model can handle touching surfaces.
>
> 3. As specified in the appendix, the object sizes in our dataset vary in a pre-defined range.
> In the appendix, we also demonstrate our model on scenes with four walls surrounding the plane.
> The results show that our model segment walls as part of the scene layout.
> To address the reviewer's concern, we run additional experiments and train our model with different point densities or added noise.
> For clarity, we submit our new results in the supplementary zip.
> The results are shown in the newly updated supplementary file Sec. 2.
>
> Hopefully, the majority of your concerns are addressed.
> Please let us know if you have further concerns.

---

> > ### Comment · Reviewer_PtZ1 · 2021-11-29
> > **Thanks for the rebuttal**
> >
> > I appreciate the authors' efforts in the rebuttal.
> >
> > The additional experiments on S3DIS dataset with real world scans are much more sensible than the original results on uniform-sized objects and clean scenes. However, the results suggest that the model hardly outperforms the simple clustering baseline Meanshift in these noisy real world scenes. I think these results do not appear convincing enough to me for the claim of effective unsupervised object centric learning.
> >
> > Besides, I find the authors' argument on uncolored point clouds slightly misleading. 3D point clouds do not necessarily have to be textureless; in fact, it is often much easier to obtain color information compared the scans themselves. It is a restriction of the setup in this paper. Of course there are use cases of a method dealing with untextured point clouds, but I find it misleading to claim 3D point clouds are more challenging because they come with no color information.
> >
> > Overall, I think the proposed method might be interesting, but do not think the currents results are convincing enough, and will keep my rating at 5.

---

### Official Review · Reviewer_yyA6 · 2021-11-03

**Correctness:** 3
**Technical Novelty And Significance:** 4
**Empirical Novelty And Significance:** 3
**Recommendation:** 6
**Confidence:** 5

**Main Review:**


Strengths
[novelty]
1. According to my knowledge, this is the first work that explores variational generative methods for scene-level 3D point cloud understanding; the learnt representation could be used for unsupervised 3D  instance segmentation, or scene decomposition, while existing methods like PointGroup[Jiang et al.] or 3D-BoNet[Yang et al.] requires ground truth labels to cluster neighborhood points or refine from the detected bounding boxes;

2. The proposed mixture chamfer loss is novel, which treats each point as a Gaussian distribution and turns the original Chamfer distance better for variational inference. It also utilizes a weighting mechanism when deciding each point’s corresponding glimpse, this formulation is critical for closed loop mask prediction;

[effectiveness]
1. The detection-inspired local voxel attention mechanism makes it possible to handle arbitrary number of objects in the 3D scene, handling well with 2-12 objects from their experiments;

2. The proposed method achieves performance that is on a par with supervised SOTA method on two synthetic scene point cloud datasets, including challenging cases like almost touch surfaces;

[completeness]
The author provides both quantitative and qualitative results, with ablation studies on reconstruction point number, voxel size, etc. They also mentioned its drawbacks like handling large objects that are much bigger than the voxel size, mixed objects inside one voxel.


Weakness
Though this work is sound overall, after going through the details carefully in the paper, here I’d like to point out several concerns/issues below. I have examined the key claims over the technical contributions, together with the experimental settings.

1. The chamfer mixture distance loss seems to be critical for the variational model training,  but the additional per-point probability modeling and bidirectional closest point finding, plus multiple glimpse proposals seem to make the loss computation very expensive, does the author have rough estimation of the increased computation compared to the original Chamfer distance loss? One quick related question, is it possible for the reconstruction to come in the form of regular voxels? In this way, regular one-to-one correspondence could be established naturally, without the expensive argmax operation, or searching for the closest pairs;

2. Current experiments are performed on synthetic complete scene point clouds, which is still quite a toy setting. On these datasets, there is strong bias that a flat and clean desk/room background is presented, would it be possible that the model simply learns to discard the flat plane when segmenting the object instead of learning good object-centric representations? A simple way to verify that is trying to tilt the point cloud data, so the desk plane would not be flat. Also I guess the proposed method might be limited when handling partial 3D point clouds, especially for the point cloud reconstruction supervision, is that expected?

3. Following the previous question, is it possible for the proposed method to test on S3DIS dataset as well? This is a common benchmark for 3D instance segmentation and doesn’t have quite challenging partial data. Even though there might be challenging cases due to various object scales, the author should be able to find a reasonable voxel size for glimpse generation.

4. The author is expected to provide more training details, like each glimpse may contain a different number of points, in the glimpse encoder and decoder, are different glimpses processed in batch or sequentially? Is there any difference for loss weights when training global VAE and per-glimpse VAE?

5. The author needs to be aware of some recent works in this direction, like this ICCV 2021 paper: Unsupervised Point Cloud Object Co-Segmentation by Co-Contrastive Learning and Mutual Attention Sampling[Yang et al.]

6. Could the author leverage more intuition about why this method could learn to develop a concept of visual entities in an unsupervised way? It is still amazing the proposed pipeline achieves such a good performance even on the toy datasets. But if the per-glimpse reconstruction generates points containing background points, the loss would not change, but then there is no way that  model could tell the object mask. And why do we need this global VAE for background when each glimpse module already handles the target  foreground objects?

Minor comments about weakness:
What is the foreground alpha in figure 2 (i) and (j). Besides the glimpse alpha and scene layout reconstruction, Could the author also show more reconstruction visualization in individual glimpses?
In the title, there is a word ‘Invariant', which I found quite interesting. What does this ‘invariant’ mean exactly?
Page 2. Section 3, 2nd paragraph should be ‘However, point cloud data is ...’



**Summary Of The Paper:**

This paper is a pioneering work that extends the success of unsupervised object-centric learning in 2D images to 3D scene point clouds. The framework could detect and segment multiple 3D objects from a 3D scene point cloud through a decompose-by-reconstruct strategy in an unsupervised manner.

Their variational training pipeline utilizes a VAE-based generative model that jointly considers the global 3D background and local individual objects, and a novel chamfer mixture loss is proposed for irregular 3D point clouds.  The method is validated on two synthetic datasets, and the performance is demonstrated to be close to a supervised SOTA method.


**Summary Of The Review:**

This paper is a valuable pioneering work on unsupervised 3D scene segmentation, with a full variational training pipeline built up for irregular 3D point cloud data.

---

> ### Author Response · Authors · 2021-11-18
> **Reply to reviewer yyA6**
>
> We thank the reviewer for the detailed feedback and the acknowledgment of our contribution as a pioneering work for object-centric learning for the 3D scene point cloud.
> Below we address the concerns.
>
> 1. Compared with training a point cloud generation model for the entire scene as a whole, the time complexity of chamfer mixture loss for foreground objects is, in fact, much lower.
> The key to reducing the time complexity is the local glimpse proposal scheme.
> When computing the chamfer mixture loss, the bidirectional closest point searching only happens within each glimpse.
> Namely, we do not need to compute the pairwise distance for the point cloud of the scene but rather for points only within each glimpse.
> Here we provide an example to illustrate the reduced magnitude of time complexity.
> Assume a scene consists of 10,000 points.
> We need to compute the distances of 50 million unique pairs of points.
> In contrast, there are around 1,000 points (10% of 10,000) in each glimpse, and 500k (1% of 50,000,000) unique pairs of points.
> Thus, with non-overlapping voxel grid cells, the total time complexity should be much lower than twice of chamfer distance.
> We believe the irregularity of point cloud is of both pros and cons.
> Indeed, voxel is a plausible way to obtain correspondences.
> But it is known that computation on voxel data tends to consume a large amount of memory.
> More importantly, under the regularity of voxel, gradient w.r.t to glimpse parameters are harder to obtain.
> One benefit of working with point cloud is that the point coordinates are real-valued and thus differentiable w.r.t coordinate transformation, including but not limited to scaling, translation, and rotation.
>
> 2. &
> 3. We follow the reviewer’s suggestion and evaluate our approach on S3DIS, a real dataset for 3D scenes.
> Our experiments mainly focus on scenes with regular object structures.
> We run additional experiments, and the results are shown in the newly updated supplementary files for clarity.
> Regarding partial observation, the main difficulty of working with partial observation (but complete supervision) is to determine the number of points in the reconstructed point cloud.
> The big difference between the number of reconstruction points and the number of input points destroys the balance between the forward and the backward loss.
>
> 4. Below, we provide more detail regarding our pipeline.
> It is expected that each glimpse contains a varying but non-zero number of points.
> Both the PointConv layer and the PointGNN layer are not constrained to the input of a fixed number of points.
> Reference of those two types of layers can be found in Sec. 3.4.
> Foreground glimpse encoder and decoder process all glimpses in parallel as each glimpse only focus on a local region.
> In our implementation, all glimpses are extracted and batched before being sent into the encoder.
> All weights are the same for global VAE and glimpse VAE.
>
> 5. We thank the reviewer for bringing this work to our attention.
> This work has a different setup to ours but is definitely related to our topics.
> While this work requires rough bounding boxes/spheres of the target objects (which already requires prior knowledge) and detects one type of object at a time, our model process the entire scene in one go.

---

> > ### Author Response · Authors · 2021-11-18
> > **Reply to reviewer yyA6 continue**
> >
> > *6. Similar to previous works on generative model-based scene decomposition in 2D or 3D [1][3], our model relies on the concept of "information bottleneck". Unsupervised object-centric learning is essentially an attempt to define objectness.
> > While rule-based unsupervised segmentation algorithms code low-level heuristics/human prior knowledge directly, generative model-based object-centric models are built on a high-level heuristic.
> > That is, the concept of "objectness" is a consequence of agents trying to model the world as accurately as possible but at the same time trying to process as little information as possible due to the limited computational power.
> >
> > To be more concrete, the term "object" should refer to matters that are highly correlated with each other.
> > Correlation means both predictability and compressibility.
> > Using VAE is a good way to enforce such high-level heuristics.
> > In fact, a VAE effectively compresses the input and tries to reconstruct it under an information budget.
> > Thus, highly correlated parts of inputs are forced to be compressed together, and the union of those parts is considered an object.
> > See beta-VAE related work [5][6] for the in-depth introduction.
> >
> > Previous works [1][2][3][4] applied this idea to environments with strong color cues.
> > While all works following this line heavily rely on color cues, the aim of this work is to show that the information bottleneck principle is general and can be applied to multiple data modalities, including point cloud.
> >
> > From the technical perspective, the segmentation performance is a consequence of the balance between multiple terms.
> > Below we discuss each term in detail.
> >
> > (1) reconstruction likelihood:
> > The forward likelihood and backward loss each have a trivial suboptimal.
> > The forward loss encourages reconstruction to be close to all input points.
> > One trivial suboptimal is to distribute reconstruction points densely and evenly across the entire space so that for any input point there exists a close reconstruction point.
> > The backward loss encourages reconstruction to be close to at least one input point.
> > One trivial suboptimal is to distribute all reconstruction points close to one input point.
> > The balance between the number of input points and reconstruction points suggests the trivial suboptimal for one loss is an extremely bad solution for the other loss.
> > Also, note that the reconstruction likelihood as a whole encourages over-segmentation.
> > More specifically, more glimpses lead to model parts that are easier to reconstruct compared with the whole object with non-trivial structure.
> >
> > (2) KL term of z_pres:
> > to counter the over-segmentation tendency, the prior of z_pres encourage glimpse rejection.
> > Note that once the glimpse is rejected with z_pres = 0, this glimpse will not contribute to the final loss anymore.
> > This design further encourages glimpse rejection.
> >
> > (3) KL term of z_what:
> > The KL term of z_what implements the information bottleneck idea.
> > While the KL term of z_pres encourages glimpse rejection, the bottleneck makes sure that one glimpse does not have enough capacity to model more than one object well.
> > Most importantly, z_what of scene layout branch makes sure that scene glimpse along cannot achieve a high-quality reconstruction of the entire scene.
> > Compared with AE, VAE allows a trade-off between reconstruction quality and information budget within a range.
> >
> > (4) KL term of z_where:
> > This term serves as a tie-breaker.
> > When multiple glimpses try to model the same object, this KL term encourages the closest glimpse to take the object.
> >
> > (5) KL term of glimpse size and the boundary loss:
> > The size of glimpses is encouraged to shrink so that each object only contains points that absolutely belong to this object.
> > That is how floors are excluded from each object.
> > To be more specific, if one part of an object (or floor) is already modeled by one glimpse well, then excluding this part from other glimpses will not decrease the likelihood at all.
> > The boundary loss changes the hard "excluding" process into its soft version.
> > Thus, a point being gradually excluded will change the likelihood or the mask in a differentiable way.
> >
> > Those terms regularized the behavior of glimpses.
> > A balance between them gives a good segmentation result.

---

> > > ### Author Response · Authors · 2021-11-18
> > > **Reply to reviewer yyA6 continue**
> > >
> > > Minor comments:
> > > In Figures 2 (i) and (j), the value of alpha is shown by colors.
> > > We use the standard "plasma" color map from the matplotlib where bright yellow is 1 and dark blue is zero.
> > > "Invariant" means that the foreground glimpse encoder and decoder operate without knowing the global coordinates of the points, which is the key to high scalability.
> > > Note that by using GNN with a fixed radius, we implicitly make the assumption that only local information is needed to detect objects.
> > > Thus, it is spatially invariant like a CNN layer on a 2D image.
> > > A CNN layer can take a 2D image with an arbitrary size, so can our foreground VAE.
> > > More glimpse visualization can also be found in the supplementary file  Sec. 3.
> > >
> > > Hopefully, the majority of your concerns are addressed.
> > > Please let us know if you have further concerns.
> > >
> > > [1] Klaus Greff, Rapha ̈el Lopez Kaufman, Rishabh Kabra, Nick Watters, Christopher Burgess, Daniel Zoran, Lo ̈ıc Matthey, Matthew M Botvinick, and Alexander Lerchner Multi-object representation learning with iterative variational inference. In ICML, 2019.
> > >
> > > [2] Paul Henderson and Christoph H. Lampert. Unsupervised object-centric video generation and decomposition in 3D. In NeurIPS, 2020.
> > >
> > > [3] Zhixuan Lin, Yi-Fu Wu, Skand Vishwanath Peri, Weihao Sun, Gautam Singh, Fei Deng, Jindong Jiang, and Sungjin Ahn. Space: Unsupervised object-oriented scene representation via spatial attention and decomposition. In ICLR, 2020.
> > >
> > > [4] Francesco Locatello, Dirk Weissenborn, Thomas Unterthiner, Aravindh Mahendran, Georg Heigold, Jakob Uszkoreit, Alexey osovitskiy, and Thomas Kipf. Object-centric learning with slot attention. In NeurIPS, 2020
> > >
> > > [5] rina Higgins, Lo ̈ıc Matthey, Arka Pal, Christopher Burgess, Xavier Glorot, Matthew M Botvinick, Shakir Mohamed, and Alexander Lerchner. beta-vae: Learning basic visual concepts with a constrained variational framework. In ICLR, 2017
> > >
> > > [6] Christopher P. Burgess, Irina Higgins, Arka Pal, Lo ̈ıc Matthey, Nick Watters, Guillaume Desjardins, and Alexander Lerchner. Understanding disentangling in beta-vae. ArXiv, abs/1804.03599, 2018.

---

> > > ### Comment · Reviewer_yyA6 · 2021-11-29
> > > **Reply to the authors' rebuttal**
> > >
> > > 1. Thanks for the authors' detailed reply! I appreciate the careful comments on 1. Chamfer mixture loss and 6. concept of "information bottleneck" and the loss design. The locally divided glimpse makes the computation affordable on common GPUs.
> > >
> > > 2. Regarding the additional results on S3DIS, both visually and quantitatively it didn't look impressive to me as also pointed out by Reviewer PtZ1. The method simply fails on long and complex objects, possibly failing at the step when determining the boundaries from each glimpse. But this is as expected, as the originally proposed pipeline is mainly designed to handle more complete and simple point clouds using the local anchor-like glimpse mechanism and reconstruction loss. It seems the main bottleneck comes from the reconstruction heads. Based on the author's pipeline, there should be a few easy fixs to tune the pipeline for real-world data, like weighting the two different items of mixture Chamfer loss, using multiple scales of the same input to deal with objects with different sizes and adding post merging, and etc.
> > >
> > > 3. When it comes to the evaluation of this paper, I still think this work well demonstrated the possibility of unsupervised learning of 'object' concept using information bottleneck principles from 3D scene point cloud data for the first time. While its application to 3D instance segmentation is not that good on real-world data, it promises future directions left to be explored. When there is no texture information and other necessary prior knowledge, it is even hard for us human to tell apart different 'objects'.
> > >
> > > Based on existing discussions I'd like to keep my original rating, but open to further opinions from other reviewers.

---

### Author Response · Authors · 2021-11-27
**Meta Response**

We would like to thank all reviewers for their feedback.
We're happy to hear that reviewers agree that our paper "is a pioneering work that extends the success of unsupervised object-centric learning in 2D images to 3D scene point clouds"(yyA6) and "is the first to extend 2D object-centric representation learning to 3D point clouds"(orM7).
A major concern raised by reviewers is the simplicity of our simulated dataset.
To demonstrate the effectiveness of our model, we follow suggestions from reviewers and conduct experiments on the S3DIS dataset.
The S3DIS dataset contains point clouds of real-world indoor scenes of varying sizes.
We report that our model achieves in mIoU Chair: 0.603, Table: 0.375, and Sofa: 0.482.
As a baseline, we also report that the Meanshift (with floors manually removed) achieves Chair:0.634, Table:0.297, Sofa:0.272.
Detailed experimental setup, quantitative results as well as qualitative visualizations can be found in the supplementary zip file.

We also test our model on the UOR dataset with varying point density or with injected noises.
The results show that the performance of our model is stable under such variations.
See supplementary zip file for details.

We would like to emphasize that, our work mainly focuses on object-centric representation learning that produces meaningful object representation with unsupervised instance segmentation as one application.

Built on the widely adopted SPAIR framework (especially for 2D images), SPAIR3D paves the way to taking advantage of the rich unsupervised techniques developed in the 2D object-centric learning community.

---

### Decision · Program_Chairs · 2022-01-20

**Decision:**

Reject

**Comment:**

This paper introduces a VAE-based generative model of 3D point-clouds inspired by SPAIR that can do unsupervised segmentation, named SPAIR3D. The model uses both global and local latent variables to encode global scene structure as well as individual objects.

The proposed model is relatively complex, but the presentation is overall clear.

Experimental results on simple synthetic datasets look promising. However, one might argue that for these simple tasks a direct application of a simpler mixture of VAEs (such as IODINE) might be sufficient, so it would be informative to make a direct comparison between these methods and/or show results on a problem clearly out of the scope of these simpler methods (e.g. with high imbalance in the point clouds).